# Orbital-selective effect of spin reorientation on the Dirac fermions in a non-charge-ordered kagome ferromagnet Fe₃Ge

Rui Lou [1,2,3,17] ✉, Liqin Zhou[4,5,17], Wenhua Song[6,17], Alexander Fedorov [1,2,3,17], Zhijun Tu[6,17], Bei Jiang [4,5], Qi Wang[6,7,8], Man Li [9], Zhonghao Liu [10], Xuezhi Chen[5,11], Oliver Rader [2], Bernd Büchner [1,12], Yujie Sun [13,14,15], Hongming Weng [4,5,16] ✉, Hechang Lei [6] ✉ & Shancai Wang [6] ✉

Kagome magnets provide a fascinating platform for the realization of correlated topological quantum phases under various magnetic ground states. However, the effect of the magnetic spin configurations on the characteristic electronic structure of the kagome-lattice layer remains elusive. Here, utilizing angle-resolved photoemission spectroscopy and density functional theory calculations, we report the spectroscopic evidence for the spin-reorientation effect of a kagome ferromagnet Fe₃Ge, which is composed solely of kagome planes. As the Fe moments cant from the $c$-axis into the $ab$ plane upon cooling, the two kinds of kagome-derived Dirac fermions respond quite differently. The one with less-dispersive bands ($k_z \sim 0$) containing the $3d_{z^2}$ orbitals evolves from gapped into nearly gapless, while the other with linear dispersions ($k_z \sim \pi$) embracing the $3d_{xz}/3d_{yz}$ components remains intact, suggesting that the effect of spin reorientation on the Dirac fermions has an orbital selectivity. Moreover, we demonstrate that there is no signature of charge order formation in Fe₃Ge, contrasting with its sibling compound FeGe, a newly established charge-density-wave kagome magnet.

The entanglement of multiple degrees of freedom, including spin, charge, and lattice, is widely believed to be responsible for the rich phase diagrams in correlated systems, where the various ordered phases being closely adjacent to each other allow the continuous tunability of the ground states[1,2]. A similar case with an intertwining between nontrivial band topology, magnetism, and symmetry-breaking states has been recently discovered in the $3d$ transition-metal-based kagome lattices. A tight-binding model on the kagome lattice yields a symmetry-protected electronic structure hosting the flat band (FB) over the entire Brillouin zone (BZ), the Dirac point (DP) at the BZ corner, and the van Hove singularities (vHSs) at the BZ boundary. A further combination of its unique lattice geometry and the intrinsic magnetism has been reported to engender a variety of novel quantum phenomena, such as the massive Dirac fermions[3] and skyrmion bubble states[4] in ferromagnetic (FM) Fe₃Sn₂; the spin-

polarized FB in antiferromagnetic (AFM) FeSn[5,6]; the magnetic Weyl fermions and large anomalous Hall effect in non-collinear AFM Mn₃Sn[7,8] and Mn₃Ge[9,10]; the topological Chern magnetism in FM TbMn₆Sn₆[11]; and the FM Weyl semimetal state in Co₃Sn₂S₂[12–14]. In addition to these magnetic compounds, the recent discovery of kagome superconductors $A$V₃Sb₅ ($A$ = K, Rb, Cs) has also expanded the family of non-magnetic kagome metals[15–25]. Significant interests have been focused on the time-reversal symmetry-breaking charge-density wave (CDW) therein[26–32], which could be triggered by the interactions between multiple vHSs close to the Fermi level ($E_F$)[33–37]. Very recently, a three-dimensional (3D) CDW order with the same 2 × 2 × 2 super-structure as $A$V₃Sb₅ was observed deep inside the AFM phase of a correlated kagome metal FeGe[38–40]. The electronic instabilities might still play a role in the formation of CDW therein, as the vHSs are found to be shifted to the vicinity of $E_F$ by the AFM order[41]. This magnetism-

induced band modification provides a precious opportunity to explore the magnetic impact on the characteristic electronic structure of the kagome lattice. However, the presence of two types of terminations in FeGe (Ge and kagome terminations[39]) makes it less conducive to designing and manipulating the band structure that is intrinsic to the kagome lattice layer by the magnetic exchange interactions. A sibling material of FeGe harboring only the kagome termination is therefore desirable. Meanwhile, a lack of close siblings of FeGe and comparative studies between them thus far also hinder the elucidation of the mechanism of charge order in FeGe.

Here, we combine angle-resolved photoemission spectroscopy (ARPES) and density functional theory (DFT) calculations to study the electronic properties and correlation effects of a kagome ferromagnet $Fe_3Ge$. Compared to the structure of FeGe, where the neighboring $Fe_3Ge$ kagome layers are well separated by the Ge honeycomb layer[42,43], the $Fe_3Ge$ compound consists only of the directly stacked $Fe_3Ge$ kagome layers, as shown in Fig. 1a. We find that the structural three-dimensionality of $Fe_3Ge$ is reflected in its electronic structure featuring 3D characters, particularly in the two kagome-derived Dirac fermions with different $k_z$ values exhibiting distinct dispersions.

Moreover, we reveal that these two Dirac fermions show different responses as the Fe moments cant from the $c$-axis into the $ab$ plane[44]— the first (DP1) formed by the less-dispersive bands ($k_z \sim 0$) evolves from gapped into nearly gapless; while the second (DP2) with the linear band dispersions ($k_z \sim \pi$) is much less affected. According to the orbital-resolved DFT calculations, the DP1 has a mixture of the $3d_{xy}/3d_{x^2-y^2}$ and $3d_{z^2}$ orbitals, and the DP2 is mainly composed of the $3d_{xy}/3d_{x^2-y^2}$ and $3d_{xz}/3d_{yz}$ orbitals. These results, taken together, indicate the orbital-selective effect of spin reorientation on the Dirac fermions. Furthermore, we find that, in contrast to FeGe[38,41], there is no signature of CDW order formation in $Fe_3Ge$. The electron correlations and the nesting of vHSs of $Fe_3Ge$ have been quantitatively examined and seriously discussed. These studies suggest the presence of orbital-selective vHSs near $E_F$ as the vital ingredient for triggering the CDW instabilities. Our work not only provides insightful information for the origin of the CDW order in magnetic kagome metals, but also establishes the first kagome system in which the mass of Dirac fermion can be controlled by the intrinsic magnetism. The high possibility of switching the Dirac electrons in $Fe_3Ge$ by external magnetic fields paves the way for new functionalities.

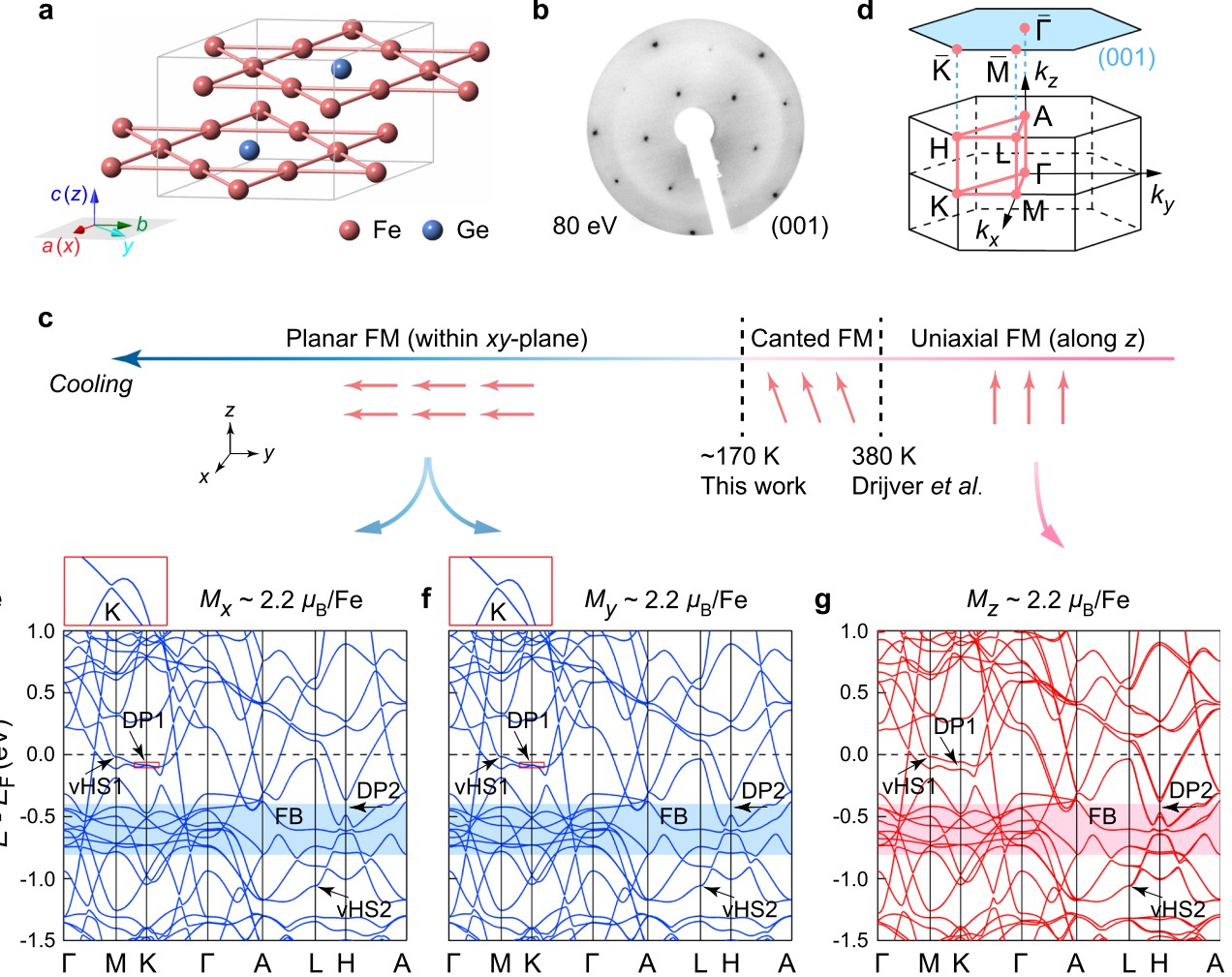

**Fig. 1 | Crystal structure, magnetism, and band calculations. a** Schematic crystal structure of $Fe_3Ge$. The inset illustrates the setup of the $x$, $y$, and $z$ axes relative to the $a$, $b$, and $c$ axes, where the $x$ and $z$ axes are along the $a$ and $c$ axes, respectively, the $y$-axis is in the $ab$ plane, at an angle of 30° to the $b$-axis. **b** LEED pattern of the treated $Fe_3Ge$-(001) surface taken with an electron energy of 80 eV at room temperature. **c** A summary of the magnetic phase transitions as a function of temperature in $Fe_3Ge$. **d** Bulk and (001)-projected BZs with the high-symmetry points. **e–g** DFT calculated band structures of $Fe_3Ge$ in the FM phase for the Fe moments ($\mu_{Fe} \approx 2.2\mu_B$) aligned along the $x$ (**e**), $y$ (**f**), and $z$ (**g**) axis, respectively. As guided by the curved arrows, one can find the correspondence between the band calculations and the temperature axis in **c**. Two sets of DPs and vHSs below $E_F$ are highlighted by the black arrows. The FB regions at around $-0.5$ eV are marked out by the blue and red shades. As indicated by the red rectangles in **e** and **f**, the insets of which show the enlarged view of the nearly gapless (-0.5 meV) DP1 at $K$ point in the planar FM state.

## Results

Fe₃Ge is isostructural to Mn₃Sn[7,8] and Mn₃Ge[9,10], crystalizing in a hexagonal structure with the $P6_3/mmc$ (No. 194) space group and belonging to the kagome lattice binary $T_mX_n$ series ($T$ = Mn, Fe, Co; $X$ = Sn, Ge; $m:n$ = 3:1, 3:2, 1:1)[5]. Like the Mn₃Ge bulk crystals[45], we found that it is almost unfeasible to obtain atomically flat surfaces for the surface-sensitive ARPES experiments by cleaving the single crystals of Fe₃Ge. We thus polished the (001) surfaces of Fe₃Ge crystals and then sputtered and annealed the polished surfaces in the vacuum[46]. The sputtering and annealing procedures were repeated several times until we observed clear low-energy electron diffraction (LEED) patterns, as shown in Fig. 1b and Supplementary Fig. S1. Such sputter-anneal treatments have little effect on the sample stoichiometry (see Supplementary Fig. S2a, b and Note 1), ensuring the reliability of our photoemission results. The treated sample surfaces have been further characterized by X-ray photoelectron spectroscopy measurements (Supplementary Fig. S2c), where the doublet structures of Ge-3$d_{3/2}$ and Ge-3$d_{5/2}$ core levels are identified, pointing to the bulk-surface splitting (see Supplementary Fig. S2d and Note 1 for quantitative fitting and detailed discussion). A uniaxial FM ordering with moments oriented along the $z$-axis was found to appear below $T_c \approx 650$ K in hexagonal Fe₃Ge[47], the magnetization directions here and hereafter are described in the Cartesian coordinates (the setup of the $x$, $y$, and $z$ axes relative to the $a$, $b$, and $c$ axes is sketched in the inset of Fig. 1a). As depicted in Fig. 1c, it was reported that the Fe moments start to cant away from the $z$-axis towards the $xy$ plane at a lower temperature of $T_{SR} \approx 380$ K and finally, a planar FM ground state is formed[44] (below $T_{planar} \sim 170$ K, which is determined by our ARPES results presented below).

Figure 1d shows the bulk and (001)-projected surface BZs of Fe₃Ge. Along the high-symmetry directions marked out in Fig. 1d (red solid lines), we plot the overall bulk band structure from DFT calculations in the FM state in Fig. 1e–g, where the preferred spin direction is along the $x$, $y$, and $z$-axis, respectively (the calculated magnetic moment of ~2.2 $\mu_B$/Fe is close to the previous experimental value of ~2.0 $\mu_B$/Fe[44]). Below $E_F$, a FB over the whole BZ and two sets of DPs and vHSs (denoted as DP1,2 and vHS1,2) are identified. We find that these kagome bands are shifted much further below $E_F$ upon entering the paramagnetic phase (Supplementary Fig. S3), similar to the case of FeGe[41]. The structural three-dimensionality of the kagome lattice gives rise to quite different electronic structures in the $k_z = 0$ and $\pi$ planes, in particular, the gapped Dirac fermions DP1 and DP2, which are formed by the less-dispersive bands and linear band dispersions, respectively. When the spin is embedded in the $xy$ plane, no noticeable difference is found between Fig. 1e, f, indicating that, in the planar FM state of Fe₃Ge, the orientation of Fe moments has a negligible effect on the overall electronic structure; interestingly, as the reorientation of spin from the $xy$ plane (Fig. 1e, f) towards the $z$-axis (Fig. 1g), one notices that, besides the small lift of band degeneracy along the $\Gamma$–$A$ and $A$–$H$ –$L$ paths (see Supplementary Note 2 and Tables S1–S3 for the interpretation), the Dirac fermions DP1 and DP2 respond quite differently, with the Dirac gap at DP1 being remarkably enhanced (from ~0.5 to ~60 meV) and the Dirac gap at DP2 being less affected (from ~120 to ~100 meV).

In order to characterize the typical kagome electronic structure in the planar FM state, we carried out the polarization- and photon-energy-dependent ARPES measurements on the (001) surfaces of Fe₃Ge. The key signatures of the kagome lattice are summarized in Fig. 2. According to the ARPES intensity map in the $hv$–$k_\parallel$ plane (Supplementary Fig. S4a), we determine that the photon energies of 125 and 146 (116) eV are respectively close to the $k_z$ values of 0 and $\pi$. In Fig. 2a, we present the Fermi surface (FS) intensity plot in the $k_z \sim 0$ plane. There exist two hole-like pockets centered at $\Gamma$ point, consistent with the DFT calculations (Fig. 2c and Supplementary Fig. S5a). The outer FS is vanishingly weak along the $\Gamma$–$K$ direction of the first BZ, while it is more visible in the second BZ due to matrix element effects. As shown

in Fig. 2b and Supplementary Fig. S4b, c, the FS mapping recorded near the $k_z \sim \pi$ plane displays a richer topology than that of the $k_z \sim 0$ plane. The most prominent feature therein is a circular electron-like pocket around $H$ point, which arises out of the kagome lattice-derived Dirac bands of the DP2. The Dirac cone structure can be directly visualized in a series of constant-energy maps in Fig. 2b. Along with the energy going below $E_F$, the circular FS contour (guided by the red circle) first shrinks into a single point at $E \approx -0.10$ eV and then expands out again. This Dirac pocket matches well with the DFT (Supplementary Fig. S5c, d) with a slight difference in the pocket size (i.e., the energy position of DP2), which can be derived from the presence of moderate electron-electron correlations in Fe₃Ge (discussed below) that are not included in the DFT. With a bandwidth renormalization factor of about 3 (Supplementary Table S4), the overall ARPES spectra could show a markedly reduced bandwidth compared with that in the DFT calculations, thus giving rise to such discrepancy. By a further comparison of the ARPES mappings and DFT FSs in $k_z \sim \pi$, one finds that the other rich topologies in ARPES are not well reproduced by the theory, in contrast to the case in $k_z \sim 0$. As discussed later, the ARPES intensity in $k_z \sim \pi$ suffers from a stronger $k_z$ broadening effect than $k_z \sim 0$, it is thus inferred that these additional spectral patterns could be the projections from other $k_z$ planes.

To further study the underlying dispersions of the DP2, we measured the near-$E_F$ ARPES spectra along the $H$–$L$–$H$ direction. As shown in Fig. 2d, e, the linear bands cross each other at about −0.10 eV to form the DP2 at $H$ point, agreeing with the evolution of Dirac fermiology in Fig. 2b. We further observe that the lower branch of the DP2 disperses down to form the vHS2 at about −0.35 eV at $L$ point, as expected from the kagome tight-binding model[48–50]. Besides this set of DP and vHS, we also identify the DP1 (at about −0.13 eV, $K$ point) and vHS1 (at about −0.11 eV, $M$ point) in the $k_z \sim 0$ plane (Fig. 2f, g). Because of the small difference in the binding energies of DP1 and vHS1, as well as the DP1 consisting of less-dispersive bands as suggested by the DFT calculations, the dispersion that connects the DP1 and vHS1 is nearly flat and almost indistinguishable from the lower branch of DP1 along the $K$–$M$ direction. Consequently, these two dispersions forming the DP1 nearly overlap with each other, exhibiting the flat spectral feature between $K$ and $M$ points, as shown in Figs. 2f and 3a. But it is noted that the separation between these two bands can still be resolved when approaching $M$ point, with one forming the vHS1 and the other merging with the $\beta$ band (Fig. 3a). Such separation near $M$ point is further demonstrated by the corresponding energy distribution curves (EDCs) presented in Supplementary Fig. S6. As for the $\Gamma$–$K$ direction, one branch of the DP1 is experimentally revealed ($\gamma$ band, Fig. 3a), and the other branch is not visible in the first BZ but in the second BZ due to matrix element effects (see Supplementary Fig. S7 and Note 3 for its presence in the second BZ). The saddle-point topology of the vHS1 can be clearly visualized by tracing the dispersions across $M$ point. In Fig. 2h, we display a series of cuts parallel to the $K$–$M$–$K$ direction (indicated by the black dashed arrows in Fig. 2a, h), the band tops of the hole-like band (guided by the red solid curve) show a minimum energy at about −0.11 eV at $M$ point, tracing out an electron-like band (red dashed curve) in the orthogonal direction ($\Gamma$–$M$ direction). In addition to the DP1 and vHS1, we further observe a broad band (at around −0.5 eV) that is nearly dispersionless over a wide range of momentum, as indicated by the blue shades in Fig. 2f, g. The energy location of this feature matches that of the kagome FB in DFT calculations (Fig. 1e, f), implying that it is of the phase-destructive-FB origin[5]. The FB nature can be better visualized by tracing the peak positions in the corresponding EDCs (Supplementary Fig. S8a, b). One can identify the FB feature also under different photon polarization (Supplementary Fig. S8c, d) and photon energy (Supplementary Fig. S8e, f).

Having fully characterized the typical band structure of the kagome lattice, we now examine the intricate effect of spin

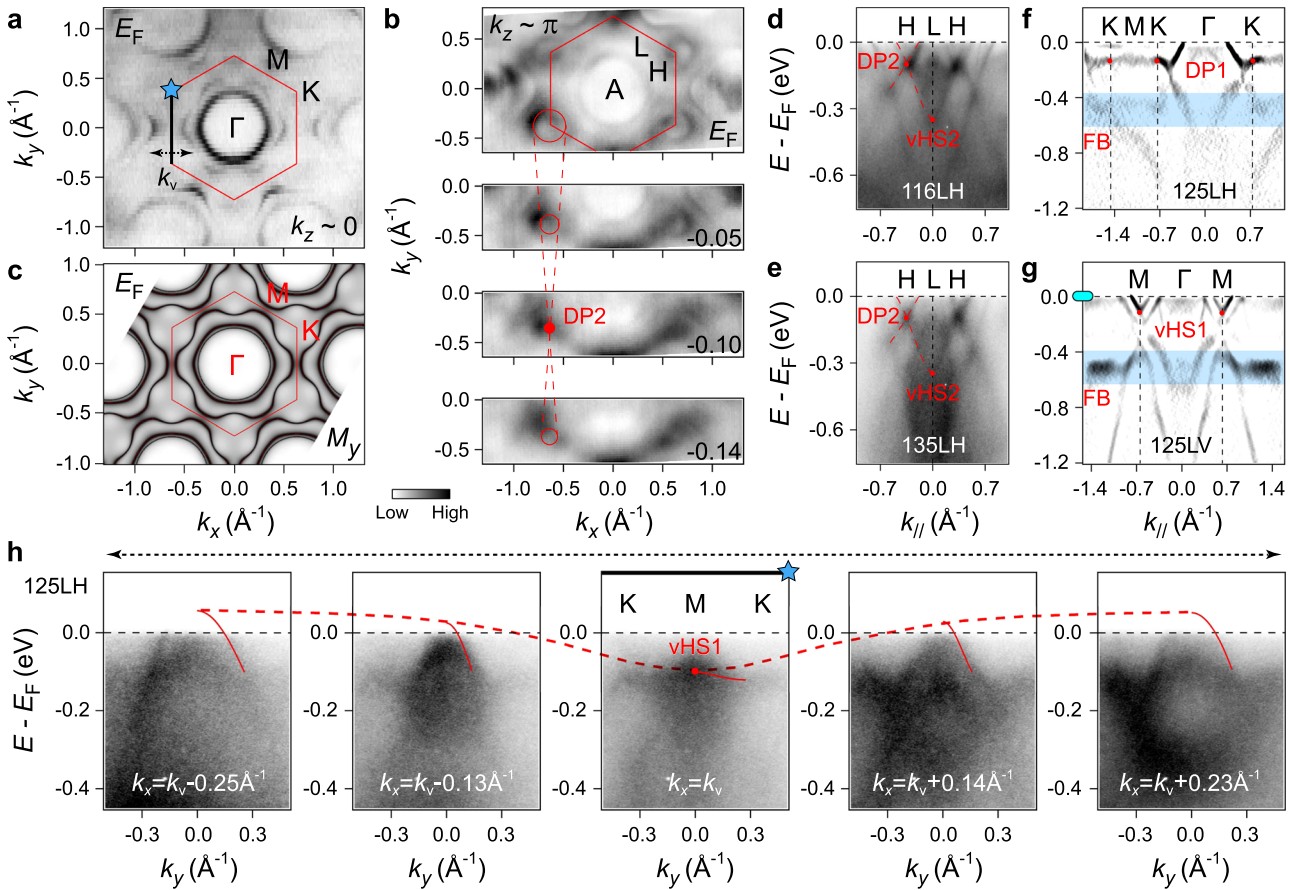

**Fig. 2 | Typical band structure of the kagome lattice. a** FS mapping of Fe₃Ge taken by the 125-eV photons (close to $k_z = 0$ plane) with linear horizontal (LH) polarization. **b,** Constant-energy ARPES intensity plots ($hv = 146$ eV, close to $k_z = \pi$ plane, LH polarization) at the energies of 0, −0.05, −0.10, and −0.14 eV, respectively. The red circles indicate the Dirac pocket at $H$ point. As guided by the red dashed lines, the Dirac pocket shrinks and reopens as the energy crosses the DP2. To avoid the complex $k_z$-projected spectra (see Supplementary Fig. S4b, c), the 116-eV photons were not utilized to study the DP2-related spectral features here and hereafter, except for the measurements in **d**, which will be explained later. **c** DFT calculated bulk FSs of Fe₃Ge in the $k_z = 0$ plane. The calculations were carried out by considering a FM moment ($\mu_{Fe} \approx 2.2\mu_B$) aligned along the $y$-axis. **d, e** ARPES intensity plots measured along the $H$–$L$–$H$ directions with the photon energies of 116 (**d**) and 135 eV (**e**), respectively (LH polarization). The red dashed curves are guides to the eye for the Dirac cone structure of the DP2 at $H$ point and its connecting to the vHS2 at $L$ point. We presented the 116-eV data in **d** for two reasons: one is to show the dispersion connecting the DP2 and vHS2, which is much clearer

than that in the 146-eV data; and the other is, more importantly, to compare with the 135-eV data in **e**. Although the 135-eV photons correspond to the $k_z$ value of $\lesssim \pi/2$ (Supplementary Fig. S4a), the ARPES spectra are analogous to that under the 116-eV photons ($k_z \sim \pi$), not only showing the existence of $k_z$ broadening but additionally indicating that the ARPES intensity in $k_z \sim \pi$ suffers from a stronger $k_z$ broadening effect than $k_z \sim 0$. For convenience, we use the BZ notation of $k_z = \pi$ plane for the 135-eV data. **f, g** Second derivative intensity plots taken along the $\Gamma$–$K$–$M$ (**f**) and $\Gamma$–$M$ (**g**) directions, respectively. The 125-eV photons with LH (**f**) and linear vertical (LV) (**g**) polarizations were utilized. The less-dispersive DP1 at $K$ point, the vHS1 at $M$ point, and the FB regions are marked out. **h** Dispersions across the vHS1. As guided by the black dashed arrows in **a** and **h**, these cuts are recorded along the $k_y$ directions at $k_x = k_v − 0.25$ Å⁻¹, $k_v − 0.13$ Å⁻¹, $k_v$, $k_v + 0.14$ Å⁻¹, and $k_v + 0.23$ Å⁻¹, respectively. $k_v$ stands for the momentum value ($k_x$ direction) of vHS1. The blue star marker and thick black line indicate the cut along the $K$–$M$–$K$ direction. The red solid and dashed curves mark out the dispersions of the vHS1 along the $k_y$ and $k_x$ directions, respectively.

reorientation on the electronic structures via the temperature-dependent ARPES measurements. Figure 3a summarizes the experimental band dispersions along the $\Gamma$–$M$–$K$–$\Gamma$ lines deep inside the planar FM state ($T = 16$ K, $hv = 125$ eV), the results match well with the DFT calculations (upper panel of Fig. 3b), in particular the near-$E_F$ bands that are marked out. The calculations suggest that the reorientation of spin mainly modifies the Dirac band gap at the DP1 (Fig. 1e–g and 3b). Since the labeled vHS1 and $\alpha$, $\beta$, $\gamma$ bands, which connect to the upper or lower branch of the DP1, might also be modified accordingly, we thus study the temperature evolutions of these features together with the DP1. As indicated by the cyan markers in Fig. 3a, we plot the corresponding EDCs (ellipse, triangle, and star markers) and momentum distribution curves (MDCs) (capsule-like marker and cyan dashed line at $E_F$) at various temperatures in Fig. 3c–f, where the temperatures are illustrated using the same colors as the labels of Fig. 3g.

As shown in Fig. 3c, when we increase the temperature from 16 to 170 K, the EDC peak position of the DP1 remains at a constant energy (about −0.13 eV), compatible with the nearly gapless nature of the DP1 in the planar FM state expected from the DFT calculations (-0.5 meV, Fig. 1e, f). Upon further warming up the sample to 210 K, the single EDC peak of the DP1 is clearly split into a double-hump structure, pointing to a gap opening of ~50 meV (see the reproducibility of the temperature evolution and the quantitative determination and analysis of the DP1 gap in Supplementary Fig. S9a–e and Note 4). Given the observation of a comparable gap value at a slightly higher temperature (Supplementary Fig. S9f, g and Note 4), the trend of the DP1 gap evolution is well supported. This remarkable change of DP1 is in line with the theoretical expectation (Fig. 1e–g) that the Dirac gap at DP1 can be tuned by the spin reorientation; accordingly, it can also be obtained that, once the temperature is increased above ~170 K, the Fe moments start to cant away from the $xy$ plane towards the $z$-axis. The

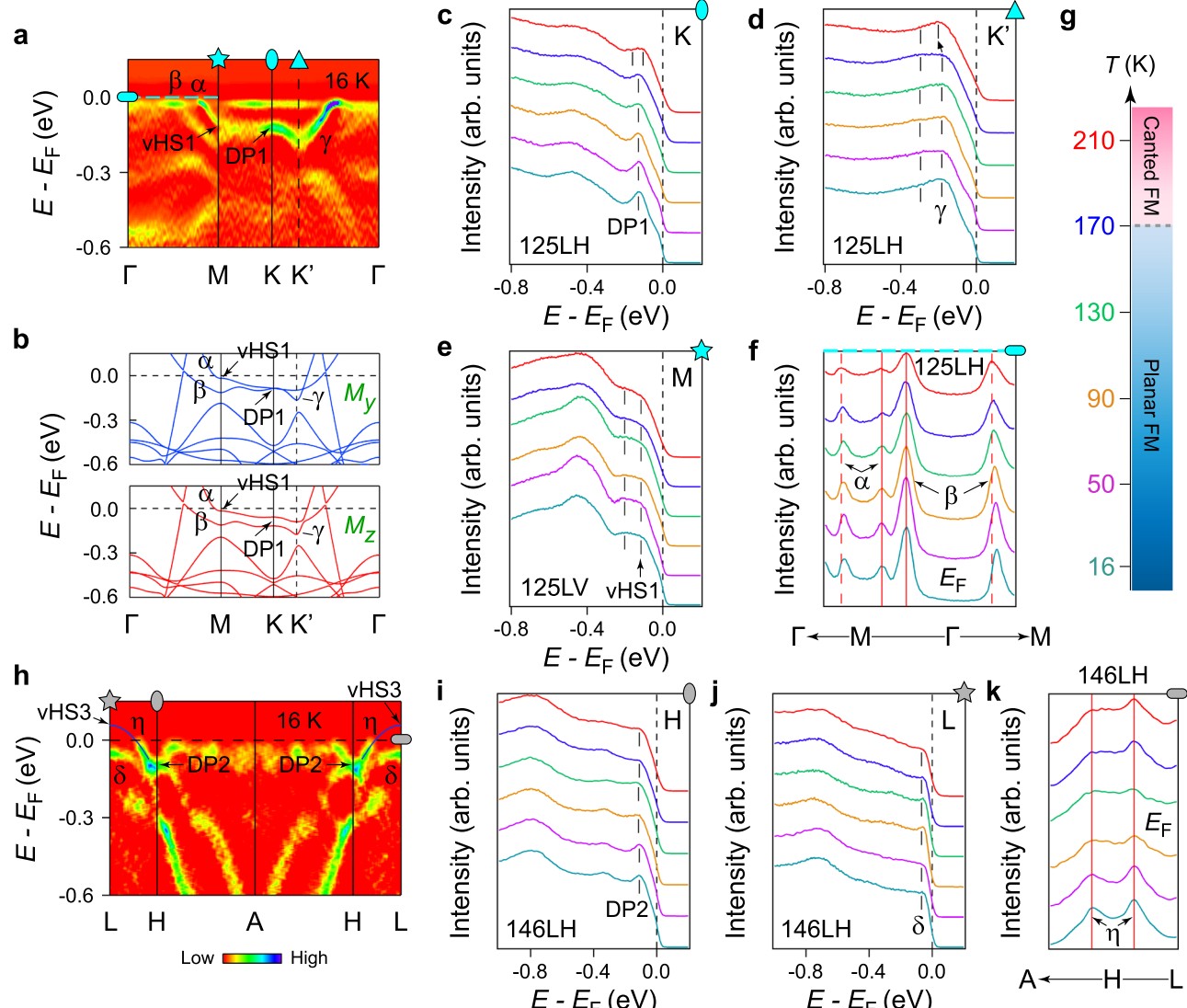

**Fig. 3 | Spectroscopic signature of the spin-reorientation effect. a** A summary of the band structures (second derivative intensity plots) of Fe₃Ge along the *Γ–M–K–Γ* lines (*hν* = 125 eV, *T* = 16 K). The cyan markers indicate the locations where the EDCs (ellipse, triangle, and star markers) and MDCs (capsule-like marker and cyan dashed line at *E*_F) in **c–f** are extracted. **b** Comparison between the calculated bands near *E*_F (*Γ–M–K–Γ* lines) of FM Fe₃Ge with the Fe moments (*μ*_Fe ≈ 2.2*μ*_B) along the *y* (upper panel) and *z* (lower panel) axes. **c–e** Temperature-dependent EDCs taken at *K* (**c**), *K′* (**d**), and *M* (**e**) points, respectively. The black dashes are extracted from peak positions. The black arrow in **d** illustrates the downward shift of the *γ* band. In **e**, the LV-polarized light was used to better reveal the *β* band bottom at *M* point. **f** Temperature-dependent MDCs along the *Γ–M* direction taken at *E*_F, as also indicated by the capsule-like marker in Fig. 2g. To better visualize the evolutions of the

MDC peaks, we align one of the *α*, *β* bands branches at various temperatures to a certain momentum position, respectively, as indicated by the red solid lines. **g** Magnetic phase diagram of Fe₃Ge based on our temperature-dependent measurements. **h** Second derivative intensity plot measured along the *A–H–L* lines (*hν* = 146 eV, *T* = 16 K). As illustrated by the blue solid curves, we use a parabolic fit to the occupied band *η* to extrapolate the location of vHS3 above *E*_F. The gray markers indicate the locations where the EDCs (ellipse and star markers) and MDCs (capsule-like marker) in **i–k** are extracted. **i, j** Temperature-dependent EDCs taken at *H* (**i**) and *L* (**j**) points, respectively. The black dashes are extracted from peak positions. **k** Temperature-dependent MDCs along the *A–H–L* direction taken at *E*_F. The red solid lines indicate the less temperature-sensitive *k*_F's of the *η* band.

spectroscopic response to the spin-reorientation effect can also be revealed at the bottom of the *γ* band, whose momentum location is denoted as *K′* point (Fig. 3a). By tracing the peak positions of the EDCs through *K′* point (Fig. 3d), one observes that the *γ* band bottom is slightly pushed down ~15 meV (highlighted by the black arrow) once the temperature is increased from 170 to 210 K. This behavior is consistent with our DFT calculations, where the downward movement of the lower branch of the DP1 upon the gap enlargement pushes the *γ* band bottom to slightly higher binding energy (Fig. 3b). Based on the spectroscopic evidence, we sketch the magnetic phase diagram of Fe₃Ge in Fig. 3g.

Figure 3e, f shows the temperature dependence of the vHS1 and *α*, *β* bands. In contrast to the DP1 and *γ* band, the bottoms of the *α* (i.e., the vHS1, at about −0.11 eV) and *β* (at about −0.20 eV) bands at *M* point are less temperature sensitive as evidenced by the EDCs in Fig. 3e. We further look into the Fermi crossings of *α* and *β* bands via the MDCs along the *Γ–M* direction taken at *E*_F. To better visualize their temperature behavior, we artificially align one of the *α*, *β* bands branches at various temperatures to a certain momentum position, respectively, as indicated by the two red solid lines in Fig. 3f. The MDC peaks from the other branches are both observed to gradually shift towards *M* point as the temperature is lowered (guided by the two red dashed lines in

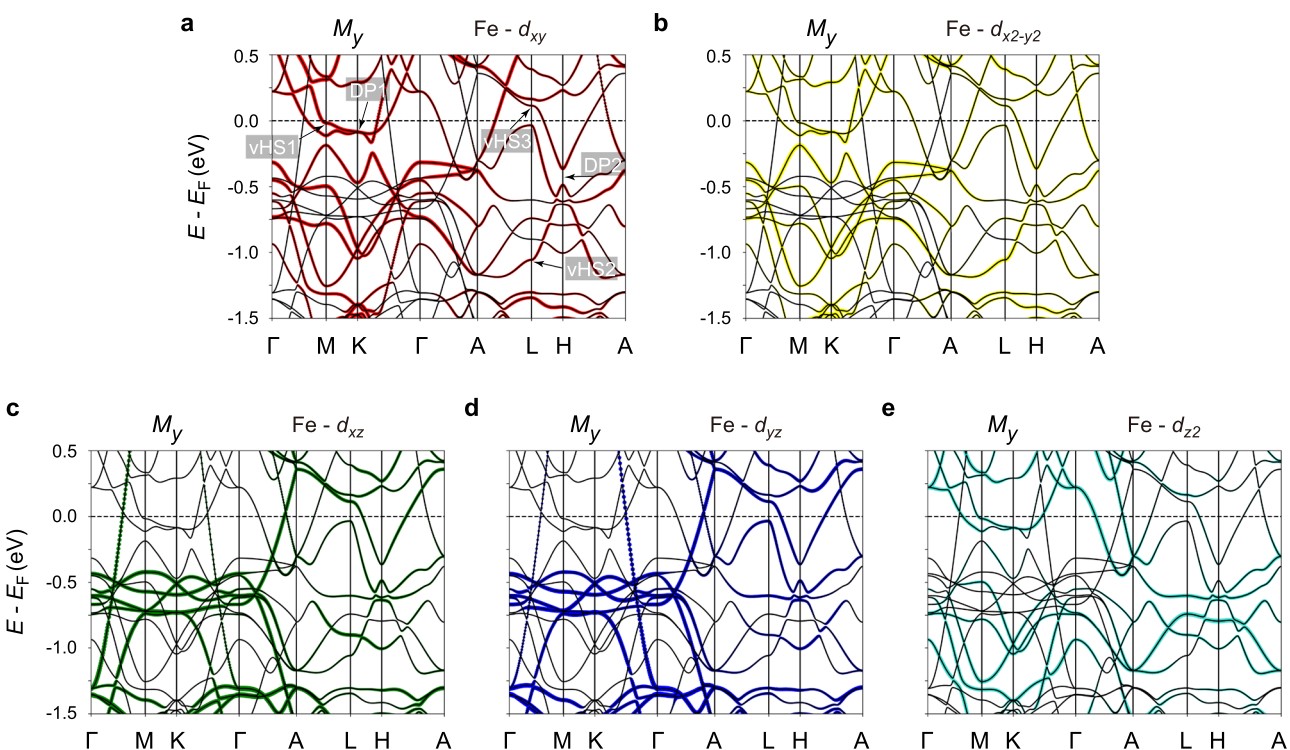

**Fig. 4 | Orbital-resolved Fe-3d band structure. a–e** DFT electronic structures of FM Fe$_3$Ge for the Fe moments ($\mu_{Fe} \approx 2.2\mu_B$) aligned along the $y$-axis with spectral weight projected onto five Fe 3$d$ orbitals, respectively. The red, yellow, green, blue, and turquoise colors represent the $d_{xy}$ (**a**), $d_{x^2-y^2}$ (**b**), $d_{xz}$ (**c**), $d_{yz}$ (**d**), and $d_{z^2}$ (**e**) orbital characters, respectively. The DP1,2 and vHS1,2 below $E_F$, as well as the vHS3 slightly above $E_F$, are marked out in **a**.

Fig. 3f). These facts taken together suggest that the Fermi velocities (the effective masses) of $\alpha$ and $\beta$ bands are increased (reduced) upon cooling, whose implications will be discussed later.

Next, we proceed to the temperature-dependent study of the electronic structures in the $k_z \sim \pi$ plane. Figure 3h displays the band dispersions along the $A-H-L$ lines ($T = 16$ K, $h\nu = 146$ eV), where the DP2 and a hole-like band top ($\delta$) are identified at $H$ and $L$ points, respectively, consistent with that in Fig. 2d, e and the DFT calculations (Fig. 1e, f). As seen in Fig. 1e–g and Supplementary Note 2, for $k_z = \pi$ plane, the spin-reorientation effect mainly induces the small band degeneracy lifting along the $A-H-L$ paths, which could be beyond the capability of our measurements. Accordingly, from the EDCs and MDCs at various temperatures, we do not see visible change in the binding energies for the DP2 (at about −0.10 eV) and $\delta$ band (at about −0.06 eV) (Fig. 3i, j), as well as the Fermi wave vector ($k_F$) for the upper branch of the DP2 (i.e., the $\eta$ band, Fig. 3k). One notices that the Dirac gap at the DP2 predicted by the calculations (Fig. 1e–g) is not identified in the experiments (Fig. 3h, i). The $k_z$ broadening effect is found to be significant in the vacuum ultraviolet ARPES, where the spectra reflect the electronic states integrated over a certain $k_z$ region of the bulk BZ[51,52]. We find that here the ARPES intensity in the $k_z \sim \pi$ plane suffers from a stronger $k_z$ broadening effect than the $k_z \sim 0$ plane, this is reflected in the following two aspects. (i) The spectra taken with 135-eV photons, which correspond to the $k_z$ value of $\lesssim \pi/2$, still show many similarities with that from the $k_z \sim \pi$ plane, as seen in Fig. 2d, e and Supplementary Fig. S4b–e. Such observations suggest that the projected electronic states from $k_z = \pi$ plane make a dominant contribution over a wide $k_z$ interval which is larger than $\pi \pm \pi/2$. (ii) The ARPES mapping and band structure in the $k_z \sim 0$ plane (Figs. 2a and 3a) are relatively simple and in good agreement with the DFT calculations (Figs. 2c and 3b), while the ARPES data in the $k_z \sim \pi$ plane (Figs. 2b and Supplementary Fig. S4b, c) show very rich spectral features and many of them are not well reproduced by DFT (Supplementary Fig. S5c, d). The distinct strength of $k_z$ broadening around $k_z \sim 0$ and $\sim \pi$ is due to the overall

bands along $k_z$ direction dispersing steeply around $\pi$ while becoming relatively gently dispersive around 0, as seen from the DFT band structures along $\Gamma-A$ direction in Fig. 1e–g. In this context, the ARPES spectra in the $k_z \sim 0$ plane are less affected by the $k_z$ broadening, ensuring that our observation of the temperature-dependent gap at the DP1 is intrinsic; whereas the Dirac cone structure of the DP2 observed in the $k_z \sim \pi$ plane actually also contains the projections from the nearby $k_z$ planes, which could host the Dirac cones with slightly different dispersions (see the additional FS pockets at $H$ point in Supplementary Fig. S4b, c). As a result, the Dirac gap at the DP2 could be smeared out by the $k_z$ projections.

To reveal the origin of the distinct responses of these two Dirac fermions (DP1 and DP2) to the spin-reorientation effect, we carried out the orbital-projected DFT calculations in the planar FM state, as presented in Fig. 4 and Supplementary Figs. S10 and S11. By comparing the orbitally-resolved Fe-3$d$ (Supplementary Fig. S11a, b) and Ge-4$p$ (Supplementary Fig. S11c, d) band structures, one finds that the energy bands around $E_F$ are dominated by the Fe-3$d$ orbitals (as also seen in the density-of-states (DOS) calculations in Supplementary Fig. S12), similar to the case of FeGe[41,53] and FeSn[5,54,55]. Furthermore, we identify that the DP1 and vHS1 are primarily associated with the Fe-3$d_{xy}$, Fe-3$d_{x^2-y^2}$, and Fe-3$d_{z^2}$ orbitals, while the DP2 and vHS2 are mainly contributed by the Fe-3$d_{xy}$, Fe-3$d_{x^2-y^2}$, Fe-3$d_{xz}$, and Fe-3$d_{yz}$ orbitals. One notices that DP2 also has a subtle contribution from the Fe-3$d_{z^2}$ character, but its orbital weight is vanishingly tiny compared to that of DP1. The Dirac fermions composed of different orbital characters have also been reported in other 3$d$ transition-metal kagome magnets[5,6,41,53–58]. Therefore, here, the orbital differentiation between the DP1 and DP2 points to a picture of orbital-selective response of the Dirac fermions to the spin-reorientation effect.

On the other hand, the behavior of the Dirac gap at DP1 resembles that of the Kane-Mele type spin-orbit coupling (SOC) gap[59], which is negligible under in-plane magnetic order while strongly enhanced under out-of-plane magnetic order. This is reminiscent of the opening

of a Chern gap for the spin-polarized Dirac fermions in kagome ferromagnet $TbMn_6Sn_6$[11]. Therein, the combination of the out-of-plane magnetization and the Kane-Mele type SOC is suggested to be responsible for the large Chern gap. Thus, one can deduce that here the SOC effect of the $3d_{z^2}$ electrons dominates the gap at the DP1 and should follow the Kane-Mele scenario, exhibiting a strong anisotropy. Very recently, by substituting Gd with Tb in kagome magnet $GdMn_6Sn_6$, Cheng et al. reported an in-plane to out-of-plane magnetic easy-axis reorientation and, accordingly, a tunable SOC gap at the Dirac crossing, which follows the Kane-Mele scenario[60]. There the authors suggested the corresponding Dirac cone originates from the Mn-$3d_{z^2}$ orbitals, which is consistent with our finding and interpretation of the data for $Fe_3Ge$. As for the SOC effects of the $3d_{xy}/3d_{x^2-y^2}$ and $3d_{xz}/3d_{yz}$ electrons, since the DP2 gap is already revealed in the non-SOC DFT calculations (Supplementary Fig. S13), we suggest that they should be negligible and contribute little to the predicted gaps at DP1 and DP2 (see detailed discussion in Supplementary Note 5). Regarding the origin of the Dirac gap at DP2, which is nearly independent of the magnetization direction, one potential source is the spin chirality, which produces a gauge flux and has been proposed to be able to open a Dirac gap independent of the spin orientation[61,62]. Future spin-resolved ARPES measurements are desired to reveal whether the dispersions of the DP2 have chiral spin textures.

Last but not least, the emergent correlated phases in kagome metals, such as the chiral CDW order, have drawn considerable attention[63,64]. Differently from the weakly electron-correlated $AV_3Sb_5$, where the CDW could be intimately related to the nesting of vHSs[26,28,32,33], in the moderately correlated FeGe, it has been proposed that not only the vHSs near $E_F$ but also the electron correlation effects should be considered in understanding the origin of the CDW instabilities[41,53,65]. But so far, it is still quite unclear which one is the major underlying factor triggering the CDW transition in the magnetic kagome system. We now examine these two aspects as well as whether the CDW develops in $Fe_3Ge$ and compare our results with those reported in FeGe.

We first study the strength of electron correlations in $Fe_3Ge$. We quantitatively compare the band features between experiments (at base temperature) and DFT calculations (planar FM state). The details are summarized in Supplementary Table S4. An overall renormalization factor of 2–3 is obtained, which is comparable with that in the previous ARPES study of FeGe[41], showing that $Fe_3Ge$ is a moderately correlated system as FeGe. The calculated band structure with renormalization can also clarify the aforementioned difference between the experiment and the calculation in the binding energy of DP2 (Supplementary Fig. S14). Then, we examine the role of vHSs. The AFM exchange splitting in FeGe brings the vHSs to the vicinity of $E_F$, and the CDW gap opening of ~20 meV is observed on the vHS bands[41]. In the case of $Fe_3Ge$, we also see that the emergence of ferromagnetism brings the afore-discussed two sets of kagome bands closer to $E_F$ (Supplementary Fig. S3). Besides the vHS1,2, one sees a third vHS connecting to the upper branch of the DP2 (denoted as vHS3, Fig. 4 and Supplementary Fig. S10). The vHS3 is located closer to $E_F$ than the vHS1 in experiments (~60 ± 20 meV above $E_F$ depending on $k_z$, a parabolic fit to the occupied band $\eta$ is used to estimate the location of vHS3, Fig. 3h). To investigate whether the vHS1 and vHS3 can induce any electronic instabilities in $Fe_3Ge$, we calculated the zero-frequency joint DOS by the autocorrelation of the experimental FSs in the $k_z \sim 0$ and $k_z \sim \pi$ planes, respectively (see details in Supplementary Note 6). As shown in Fig. 5a, the resulting joint DOS ($k_z \sim 0$) exhibits no peaks at $M$ points, indicating the absence of nesting between the vHS1. In contrast, the autocorrelation map in Fig. 5b ($k_z \sim \pi$) shows peaks at $L$ points (the anisotropic amplitudes between the nominally equivalent $L$ points might arise from ARPES matrix elements), implying that the vHS3 is nested by a wave vector of $(\pi, 0)$. Considering the orbital differentiation between the vHS1 ($3d_{z^2}$) and vHS3 ($3d_{xz}/3d_{yz}$) (Fig. 4 and

Supplementary Fig. S10), we suggest that the contribution of vHSs to the FS nesting is most likely orbital dependent in $Fe_3Ge$, with those containing the $3d_{xz}/3d_{yz}$ orbitals being dominant. This is compatible with the recent findings in FeGe[41] and $CsV_3Sb_5$[66] that the FS contours associated with the $3d_{xz}/3d_{yz}$ vHSs provide a better nesting condition. Despite the existence of nesting between vHS3 around $k_z \sim \pi$, in contrast to the quasi-two-dimensional (quasi-2D) electronic structures of FeGe[38,41] and $AV_3Sb_5$[66], the 3D nature of $Fe_3Ge$ could give rise to vHS3 being near $E_F$ only in a small range of $k_z$, rendering this in-plane nesting not sufficient to cause the charge fluctuations on the entire 3D FS. As a result, the nesting of vHSs in $Fe_3Ge$ is not sufficiently strong to induce the electronic instabilities. This is in stark contrast to the FeGe case, where there exists strong vHS nesting, as evidenced by the presence of both in-plane and out-of-plane vHS nesting and the opening of the CDW gap on vHS bands[38,41].

Consistently, by plotting the temperature-dependent EDCs and symmetrized EDCs taken at $k_F$'s of the vHS1 and vHS3 bands (i.e., the $\alpha$ and $\eta$ bands, respectively), no signature of CDW gap opening is observed around $E_F$, as shown in Fig. 5c–f. The aforementioned enhancement of Fermi velocities for the $\alpha$ and $\beta$ bands upon cooling also implies that there is no CDW gap at $E_F$ because their evolutions are incompatible with that of the Bogoliubov quasiparticles with back-bending dispersions around $k_F$. The absence of charge ordering is further evidenced by the electrical resistivity measurements in Fig. 5g, h, where $Fe_3Ge$ shows a metallic behavior without anomalies.

## Discussion

These comprehensive studies regarding the CDW order have the following implications: given that the moderate electron correlations are present in both the non-charge-ordered $Fe_3Ge$ and the charge-ordered FeGe while the nesting of vHSs is much stronger in the latter compound[38,41], it is thus most likely that the electronic instabilities induced by the strong vHS nesting are the dominant factor for the formation of CDW order in a magnetic kagome lattice; and accordingly, the quasi-2D structural characteristics, which prepares the seedbed for the nesting of vHSs, is probably a prerequisite for the CDW transition to take place. Besides, we also deduce that, in the charge-ordered kagome magnets, there most likely exists the cooperative interplay between the electron correlations, electronic instabilities, and probably also electron-phonon coupling (a kink structure was observed on the vHS band in FeGe[41] while not seen here in $Fe_3Ge$), which could be important not only in driving the CDW but also in determining the unconventional behaviors of the CDW[67,68].

In summary, we have unequivocally revealed the orbital-selective effect of spin reorientation on the Dirac fermions in the non-charge-ordered kagome ferromagnet $Fe_3Ge$. As the reorientation of spin from the $c$-axis towards the $ab$ plane, the less-dispersive Dirac fermion of the $3d_{xy}/3d_{x^2-y^2}$ and $3d_{z^2}$ orbital characters evolves from gapped into nearly gapless, following the Kane-Mele scenario, while the linearly dispersing Dirac cone of the $3d_{xy}/3d_{x^2-y^2}$ and $3d_{xz}/3d_{yz}$ components remains unchanged, pointing to an unusual source of its Dirac gap. Furthermore, our comprehensive studies of the absence of charge ordering in $Fe_3Ge$ clearly suggest an essential role of the orbital-selective vHSs near $E_F$ in triggering the CDW transition in a magnetic kagome lattice.

The magnetic modification of the gapped Dirac fermion observed here is similar to the momentum-dependent Zeeman energy shift of the massive Dirac band in kagome magnet $YMn_6Sn_6$ driven by a magnetic field[69]. Therein, the field-induced changes of the Dirac band point to a momentum-dependent Landé $g$ factor. In the case of $Fe_3Ge$, given that the band dispersions of the Dirac fermion can already be manipulated by spontaneous magnetization, an effective $g$ factor with even more exotic behaviors, like the stronger momentum dependence, might be expected. In this context, an external magnetic field may further lead to unusual Zeeman energy shifts and band

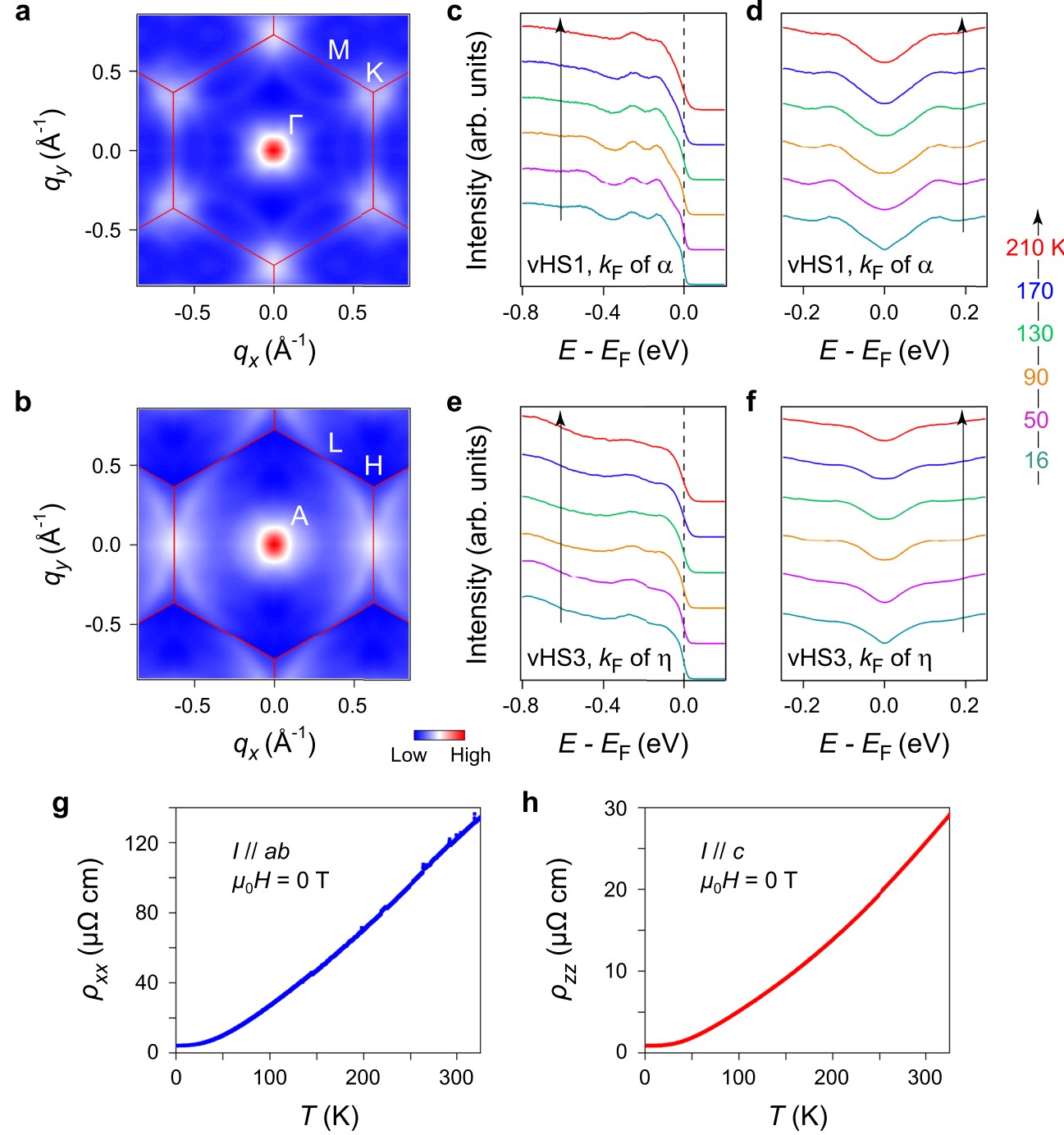

**Fig. 5 | Absence of charge ordering. a** 2D joint DOS results from the experimental FS of $Fe_3Ge$ in the $k_z$ - 0 plane. **b** Same as **a** from the experimental FS in the $k_z$ - $\pi$ plane. **c** Temperature-dependent EDCs ($h\nu = 125$ eV, LH polarization) taken at $k_F$ of the vHS1 band (i.e., the $\alpha$ band). **d** Corresponding symmetrized EDCs with respect to $E_F$ as a function of temperature. **e** Temperature-dependent EDCs ($h\nu = 146$ eV, LH polarization) taken at $k_F$ of the vHS3 band (i.e., the $\eta$ band). **f** Corresponding symmetrized EDCs with respect to $E_F$ as a function of temperature. In **c**–**f**, the temperatures are illustrated using the same colors as the labels of the rightmost diagram; the black arrows indicate the increasing temperature. **g** Temperature dependence of the resistivity $\rho_{xx}$ of $Fe_3Ge$ at $\mu_0 H = 0$ T, where $I$ is parallel to the $ab$ plane. **h** Same as **g** for the resistivity $\rho_{zz}$, where $I$ is parallel to the $c$-axis.

modifications in $Fe_3Ge$, enabling the fine tunability of the kagome electronic structure towards realizing more unconventional ground states.

## Methods

### Sample synthesis

Single crystals of $Fe_3Ge$ were grown by chemical vapor transport method. Iron powder and germane powder were mixed in a ratio of 7:3,

then were putted into evacuated fused quartz ampoule with $I_2$ as a transport agent. The transport reaction was carried out in a two-zone furnace with a temperature gradient of 1075 to 1175 K for 2 weeks. After reaction, the single crystals of $Fe_3Ge$ can be obtained.

### ARPES measurements

ARPES measurements were performed using the $1^2$-ARPES end station of UE-112-PGM2 beamline at Helmholtz Zentrum Berlin BESSY-II light

source. The energy and angular resolutions were set to better than 5 meV and 0.1°, respectively. During the experiments, the sample temperature was kept at 16 K if not specified otherwise, and the vacuum conditions were maintained better than $6 \times 10^{-11}$ Torr. In order to obtain atomically flat surfaces for the ARPES measurements, we polished the (001) surfaces of $Fe_3Ge$ single crystals, and then repeatedly sputtered and annealed the samples using an argon ion source and the electron beam, respectively, until sharp LEED patterns appeared.

## Band structure calculations

The electronic structure calculations were performed by the Vienna ab initio Simulation package with the projector augmented-wave formalism based on the DFT[70,71]. The generalized gradient approximation with the Perdew–Burke–Ernzerhof type was adopted as the exchange-correlation functional[72]. The cutoff energy for the plane-wave basis was set to be 500 eV and the local magnetic moment on Fe atoms was along the $x$, $y$, and $z$-axis for the FM state, respectively. An $10 \times 10 \times 12$ $\Gamma$-centered mesh was implemented for the BZ integral sampling. To get the tight-binding model Hamiltonian, we used the wannier90 package to obtain the maximally localized Wannier functions with Fe $p$ and $d$ orbitals[73]. The FSs of $Fe_3Ge$ were calculated by using the WannierTools software package[74]. The SOC effect was taken into account in the calculations if not specified otherwise.

## Data availability

All data needed to evaluate the conclusions in the paper are present in the paper and the Supplementary Information file. All raw data generated during the current study are available from the corresponding authors upon request.

## Code availability

The computer codes used for the band structure calculations in this study are available from the corresponding authors upon request.

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

## Acknowledgements

This work was supported by the Deutsche Forschungsgemeinschaft under Grant SFB 1143 (project C04), the Würzburg-Dresden Cluster of Excellence on Complexity and Topology in Quantum Matter—*ct.qmat* (EXC 2147, project ID 390858490), the National Natural Science Foundation of China (Grant No. 12274455), and the Ministry of Science and Technology of China (Grant No. 2022YFA1403800). Z.H.L. acknowledges the support of the National Key R&D Program of China (Grant No. 2022YFB3608000), the National Natural Science Foundation of China (Grant No. 12222413), the Natural Science Foundation of Shanghai (Grants No. 23ZR1482200 and No. 22ZR1473300), the funding of Ningbo Yongjiang Talent Program, the Natural Science Foundation of Ningbo, and Ningbo University. B.B. acknowledges the support from the BMBF via project UKRATOP. H.C.L. was supported by the National Natural Science Foundation of China (Grant No. 12274459), the Ministry of Science and Technology of China (Grant No. 2023YFA1406500), and the Beijing Natural Science Foundation (Grant No. Z200005).

## Author contributions

R.L., H.M.W., H.C.L., and S.C.W. conceived and supervised the project. Z.J.T., Q.W., and H.C.L. synthesized the single crystals and performed electrical transport measurements. B.J., Y.J.S., R.L., and A.F. processed the sample surfaces. R.L., A.F., W.H.S., M.L., Z.H.L., X.Z.C., O.R., and B.B. conducted ARPES measurements. L.Q.Z. and H.M.W. performed band structure calculations. R.L., H.M.W., H.C.L., and S.C.W. analyzed the experimental data. R.L. wrote the paper with input from all authors.

## Funding

## Competing interests

The authors declare no competing interests.

## Additional information

[1]Leibniz Institute for Solid State and Materials Research, IFW Dresden, 01069 Dresden, Sachsen, Germany. [2]Helmholtz-Zentrum Berlin für Materialien und Energie, Albert-Einstein-Straße 15, 12489 Berlin, Germany. [3]Joint Laboratory "Functional Quantum Materials" at BESSY II, 12489 Berlin, Germany. [4]Beijing National Laboratory for Condensed Matter Physics and Institute of Physics, Chinese Academy of Sciences, Beijing 100190, China. [5]University of Chinese Academy of Sciences, Beijing 100049, China. [6]Department of Physics, Key Laboratory of Quantum State Construction and Manipulation (Ministry of Education), and Beijing Key Laboratory of Opto-electronic Functional Materials & Micro-nano Devices, Renmin University of China, Beijing 100872, China. [7]School of Physical Science and Technology, ShanghaiTech University, Shanghai 201210, China. [8]ShanghaiTech Laboratory for Topological Physics, ShanghaiTech University, Shanghai 201210, China. [9]School of Information Network Security, People's Public Security University of China, Beijing 100038, China. [10]Institute of High-Pressure Physics and School of Physical Science and Technology, Ningbo University, Ningbo 315211, China. [11]Shanghai Institute of Applied Physics, Chinese Academy of Sciences, Shanghai 201800, China. [12]Institute of Solid State and Materials Physics, TU Dresden, 01062 Dresden, Sachsen, Germany. [13]Department of Physics and Guangdong Basic Research Center of Excellence for Quantum Science, Southern University of Science and Technology (SUSTech), Shenzhen 518055, China. [14]Quantum Science Center of Guangdong-Hong Kong-Macao Greater Bay Area (Guangdong), Shenzhen 518045, China. [15]Institute of Advanced Science Facilities, Shenzhen, Guangdong 518107, China. [16]Songshan Lake Materials Laboratory, Dongguan, Guangdong 523808, China. [17]These authors contributed equally: Rui Lou, Liqin Zhou, Wenhua Song, Alexander Fedorov, Zhijun Tu. ✉e-mail: lourui09@gmail.com; hmweng@iphy.ac.cn; hlei@ruc.edu.cn; scw@ruc.edu.cn

