## [Transparent Peer Review file · Nature Communications]

Orbital-selective effect of spin reorientation on the Dirac fermions in a non-charge-ordered kagome ferromagnet Fe₃Ge

Corresponding Author: Dr Rui Lou

Version 0:

Reviewer comments:

Reviewer #1

(Remarks to the Author)

In the article "Orbital-selective effect of spin reorientation on the Dirac fermions in a three-dimensional kagome ferromagnet Fe₃Ge", R. Lou et al. report on the effects spin reorientation has on the electronic structure in a three-dimensional (3D) kagome ferromagnet Fe₃Ge. The system is investigated with a combination of angle-resolved photoemission spectroscopy (ARPES) and density functional theory (DFT), with supporting characterization done by low-energy electron diffraction (LEED), and electrical resistivity measurements. Results are also compared to the known literature results on FeGe, a novel charge-density wave (CDW) kagome metal in order to explain absence of CDW in Fe₃Ge, proposing that van Hove singularities near the Fermi level play an important part in driving the CDW in Kagome systems.

While the presented results are generally interesting, the article is in my opinion more suitable for a more specialized journal. In particular, I believe it would be beneficial that authors presents at least some of their data on CDW in the main body of the article to go with their detailed discussion, and not only have it in the supplementary material, as some of the images are relevant for understanding the text. My further comments and suggestions can be found below:

1. Samples are prepared by polishing, and then series of sputtering and annealing cycles. Are the authors sure that there is no change in the stoichiometry following the procedure? It is not uncommon for some species to be sputtered away faster than others.
2. Related to question 1, and presented XPS data in Figure S2. I am surprised by a large difference in binding energy of bulk and surface component, as well as the difference in their intensity. Are authors sure that these are indeed bulk and surface components, and not Ge in different chemical environment? Furthermore, it would be nice to see these components fit.
3. Regarding the choice of photon energy: While 125 eV and 146 eV are indeed close to k_z values of 0 and π , is there a particular reason that measurements were done in vicinity of these planes, and not exactly at them?
4. Often, data at different photon energies is presented in the figures (matching approximately $\pi/2$ plane), but no particular reason is given for this (for example Figure 2). Can authors explain the choice?
5. Can authors elaborate on how measurements with different photon polarization and photon energy show that flat band is intrinsic to the kagome plane?
6. Am I correct to assume that the "nearly gapless nature of DP1 (0.5 meV)" is determined from the DFT calculations? The experimental energy resolution is 5 meV, and this is for sure too small gap to be resolved.
7. Page 10, regarding canting of the Fe moments: Authors state that change of band gap at the DP1 indicates that Fe moments start to cant away from the xy plane towards the z axis around 170 K. Is this correct? Based on the Figure 1, Fe moments are already canted at 380 K, and below 170 K they should be in the xy plane, and not canting towards z axis.
8. While k_z broadening is a serious issue when it comes to ARPES measurements of 3D materials with VUV light, I am wondering if authors have an explanation why the broadening seems to be significant around $k_z=\pi$ and not around $k_z=0$?
9. Discussion of the CDW states heavily relies on the data provided in the supplementary information, and is not so easy to follow without it.
10. Figure 1: I am not sure that I am able to see any difference between figures e) and f). I would recommend that authors either highlight some differences, or at least clarify in the text whether there are any. Furthermore, while the inset zooming in on the DP1 is helpful in seeing that there is crossing, no gap is visible in it (and I assume this is where we expect DP1 to have 0.5 meV gap).

11. Figure 2: Is there a particular reason that k-range in subfigures f) and g) is different? It seems that the only difference is that data in f) is measured with vertically polarized light and in g) with horizontally polarized light, so the k-range should also be the same for the two.
12. Figure 2: While it is possible to see what bands are used to fit the parabola at $k_x = k_y - 0.25\text{\AA}^{-1}$, it is not possible to see what band is used at $k_x = k_y - 0.13\text{\AA}^{-1}$. Also, what is the nature of the more intense, hole-like band visible at these cuts? Finally, it is not clear to me what k_y actually stands for.
13. Figure 3: I am slightly confused about the image in 3f). It has the same oval shape that was used in 3a) to indicate where the MDC is taken. Is this indeed along Γ -M- Γ -M direction, or the direction it is indicated in 3a)? Is it the former, it would be nice to also see ARPES intensity map in this direction.
14. Figure 4: It might be more informative to have larger images of orbital nature around the points of interest (DP1, DP2 vHS etc), as it is not possible to see some of the orbital contributions without zooming in quite a lot the image.
15. Figure 4: Regarding orbital contribution for vHS1, it seems like it has contribution not only from Fe-3d_{xy} and Fe-3d_{x²-y²}, but also Fe-3d_{z²} orbitals. Again, even when zooming in a lot, it is hard to really see what contributes most.

Reviewer #2

(Remarks to the Author)

Materials containing kagome lattice have attracted significant scientific interest in recent years owing to their unique electronic structures, topological properties, magnetism and superconductivity. In this manuscript, the authors studied the band structure of a kagome ferromagnet Fe₃Ge with angle-resolved photoemission spectroscopy (ARPES) and density functional theory calculations. Due to the change of temperature-induced magnetic moment direction in this system, they can capture the effect of spin-reorientation on the band structure through the ARPES experiments. The main result of the article is to track the band structures on different k_z planes, indicating that spin reorientation on Dirac fermions has orbital selectivity. However, Dirac points are not rare and appear in various systems, including honeycomb, triangular and kagome lattices, etc. In addition, the strong three-dimensional nature of Fe₃Ge makes the band structure very complex and the kagome band characteristics are not obvious compared with its sibling compound FeGe. The Dirac points studied in this manuscript are difficult to identify both theoretically and experimentally. I believe that this manuscript lacks significant findings and sufficient novelty to be published in Nature Communications. The detailed concerns and reasons are listed as follows:

1. The authors claim that "The frustrated lattice geometry of a kagome lattice can give rise to the unique electronic structure embracing the flat band..." on line 6 of page 4 in the main text. It is misleading because the frustrated lattice geometry can bring the frustrated magnetism, but is not the reason for the kagome band structure.
2. The last sentence on page 8 states that the flat band near DP1 and vHS1 is not attributed to destructive interference. The author should clarify the underlying reasons for this observation.
3. How to explain the small lift of band degeneracy in paramagnetic state along L-H-A paths in Fig. S3a? This does not appear in the ferromagnetic bands in Figs. 1e-f.
4. The Fermi surfaces on the k_z= π plane in Fig. 2b and Fig. S4 show very rich spectral patterns, which are not consistent with calculated band structures in Fig. 1. For example, three Fermi surface pockets cross the A-H path appear in Fig. S4b, while only two bands intersect with the Fermi level in Figs. 1e and 1f. It is necessary to calculate the Fermi surface on the k_z= π plane and compare it with the experimental results.
5. The energy gap at DP2 in the calculated band structure was not observed in the experiment, which is attributed to the k_z broadening effect. However, DP2 appears at different energies of about E-E_F=-0.1eV and -0.5eV in the calculated and experimental band structures, respectively. This obvious discrepancy should be explained.
6. It seems that only one energy band passes through the DP1 in the experimental band dispersion of Fig. 3a. If there is no separated bands near DP1, how can we experimentally confirm that it is a Dirac point?
7. The authors claim that DP1 is mainly contributed by a combination of the Fe-d_{xy} and Fe-d_{x²-y²} orbitals, while DP2 is primarily associated with the Fe-d_{x²-y²} orbital. However, in Fig. 4a, the Fe-d_{xy} orbital colored with red dots also seems to dramatically contribute to the DP2. This directly affects the main conclusion of this manuscript, which is that the effect of spin reorientation on Dirac fermions has orbital selectivity. To clarify this point, electronic band structures weighted by projected different Fe orbitals should be plotted separately.

Reviewer #3

(Remarks to the Author)

This work focuses on the angle-resolved-photoemission study of the Kagome ferromagnet Fe₃Ge, reporting an orbital-selective effect of the temperature-dependent spin reorientation on the Dirac fermions. The data presented in the Main text and Supplementary are of high quality and comprehensive, supporting the Authors' main conclusions. I believe the results presented here represent an intriguing advancement in the understanding of the physics of these relatively novel class of

correlated materials; therefore, I recommend the manuscript for publication in Nature Communications.

However, I do have few comments I would like the Authors' to address to strengthen the stated conclusions:

- One of the main results is the identification of a two-peak structure in the EDC of DP1 at 210K, presented in Fig.3c and S7. However, it is based only on one temperature data point (210K). Are there additional measurements taken at similar or higher temperatures which can confirm the trend?
- On the same topic, Fig. S7 shows the result of the EDC fit with two Gaussians, along the individual components. I wonder what a similar fit with a single Gaussian component would look like, and how it would compare to the 2-peaks one in terms of Chi-squares or residuals. I think an argument like this would greatly reinforce the Authors' statement.
- The Authors' attribute the lack of DP2 gap in the experiment to k_z projection broadening. How could be excluded that the same effect is taking place at DP1, affecting the probed dispersion?

Version 1:

Reviewer comments:

Reviewer #1

(Remarks to the Author)

The resubmitted draft by R. Lou et al. of the "Orbital-selective effect of spin reorientation on the Dirac fermions in a non-charge-ordered kagome ferromagnet Fe₃Ge" answers majority of my questions (not all), and before I can recommend publication in Nature Communications, I would like to have remaining questions answered and clarified.

1. Abstract states "Moreover, we present detailed comparative studies between the non-charge-ordered Fe₃Ge and its sibling compound FeGe, a newly established charge-density-wave kagome magnet, our results strongly support that the orbital-selective van Hove singularities near the Fermi level play an indispensable part in driving the novel charge order on a magnetic kagome lattice." I find this slightly misleading and would suggest rephrasing it. The presented work is done on solely Fe₃Ge, and FeGe comparison is fairly brief. Furthermore, comparison is done in comparison with various publications on FeGe, and no FeGe data is presented here.

2. Page 10: As for the Γ -K direction, one branch of the DP1 is experimentally revealed (band, Fig. 3a) while the other branch is not visible in ARPES due to matrix element effects. Authors state that they did polarization dependent measurements, and a fair amount of data in LV polarisation is presented in the supplementary information. I would like to know if they see the second band in LV polarisation, as this is not mentioned in either the main text nor in the supplementary information? If they do see the band this should be shown at least in the supplementary data. If they do not see it with LV, reasons for this need to be clarified, as so far it is only stated that band is not visible with LH due to matrix polarisation effects, which leads one to believe it should be visible with LV.

3. Regarding k_z smearing: Caption and data for figure 2, and Supplementary information figure S4. When looking at S4 data for 116 eV vs 135 eV, one can see several differences in the electronic structure. Can the authors explain what is the reason that measurements were not performed at 116 eV, but were performed at 135 eV? I would really like to know explanation for this, as Figure 3h, 16 K measurements whose data is furthermore taken at 146 eV (which is actually at $k_z = \pi$, and not 135 eV (or 116 eV).

4. Supplementary figure S6. I am not sure that I clearly see that there are 2 peaks close to the M point. Was this obtained via fitting or?

Reviewer #2

(Remarks to the Author)

I appreciate author's revisions to the manuscript. However, the revised manuscript still lacks enough novelties and interesting results. Compared to numerous systems featuring Dirac points, the current study on kagome Fe₃Ge does not appear particularly compelling. The 3D structural nature, complex band structure, and the deep energy levels of Dirac points of Fe₃Ge hinder further manipulation of these Dirac points and the observation of related quantum effects. Additionally, several other issues in the manuscript are listed below. With revisions, the manuscript could be suitable for submission to a specialized journal.

Comment 1 (Previous comment 5): The revised manuscript still does not explain the significant difference in energy levels of DP2 observed in the calculated and experimental band structures. Band renormalization from electronic correlation may account for such a discrepancy. It may be necessary to employ many-body computations such as DMFT for a more comprehensive analysis of the band structure.

Comment 2: The revised manuscript includes more discussions and Figure 5 regarding the CDW phase. In this section, by comparing between CsV₃Sb₅ and FeGe materials, the authors directly conclude on page 16 that "the vHS1 near EF arises mainly from the 3dxy/3dx²-y² and 3dz² orbital characters (Fig. 4 and Supplementary Fig. S9), it thus would not induce the

electronic instabilities in Fe₃Ge." It appears to imply that only vHS contributions from dxz/yz orbitals can lead to electronic instabilities in kagome materials. The authors should clarify this point further.

Comment 3: With both vHS1 mainly from the 3dxy/x²-y²/z² orbital characters, and vHS3 mainly from dxz/yz components near EF, Fe₃Ge does not exhibit CDW order. The main conclusion of this paper regarding the presence of orbital-selective vHSs near EF as the vital ingredient for triggering the CDW transition is quite confusing.

Reviewer #3

(Remarks to the Author)

I commend the Authors for their efforts in addressing all the reviewers' comments. All the concerns I expressed in my previous report have been satisfactorily addressed: a clear comparison between two- and one-Gaussian fit of the DP1 spectral feature is now presented and discussed in Fig.S8 and 'Supplementary Note 3', along with additional data acquired at higher temperature of 220K. I therefore recommend the manuscript for publication in Nature Communications.

Version 2:

Reviewer comments:

Reviewer #1

(Remarks to the Author)

While resubmitted draft by R. Lou et al. of the "Orbital-selective effect of spin reorientation on the Dirac fermions in a non-charge-ordered kagome ferromagnet Fe₃Ge" answers majority of my questions, I would like a further clarification for my question 3 before I can recommend this article for publication in Nature Communications.

While the authors explained and clarified well their assessment of an increased the k^z smearing at $k^z = \pi$ (116 eV, 146 eV) compared to $k^z = 0$ (125 eV), as well as reasons for (not) using $k^z = \pi/2$ data (135 eV), it is still not fully clear to me why the authors present in different figures measurements at different $k^z = \pi$ values. Namely, in figure 2b, we see data taken at 146 eV ($k^z = \pi$), while in the figure 2d we have data taken at $h\nu = 116$ eV, which is also $k^z = \pi$ based on the figure S4. However, 2D curvature and EDCs in figure 3 h,i, j and k are then taken at 146 eV. The authors state in their response (to my question 3) that the 116 eV data are in general more intense than 146 eV mappings, so I am wondering what is the reason that not all of the $k^z = \pi$ data is taken at this energy. In particular looking at the figure 2b and figure S4, 116 eV is indeed more intense and sharper, but this data is not used in the main text to show/indicate the Dirac pocket at H.

Reviewer #2

(Remarks to the Author)

I read carefully the revised manuscript as well as authors' replies, and found that most of my questions and concerns are answered. Considering the great efforts made by authors in two-round reviewing, I would give my support for publication of the revised form in NC.

Version 3:

Reviewer comments:

Reviewer #1

(Remarks to the Author)

I thank the authors for their response. All of my concerns have been addressed, and I recommend this article to be published in Nature Communications.

Reviewers' comments:

Reviewer #1 (Remarks to the Author):

In the article “Orbital-selective effect of spin reorientation on the Dirac fermions in a three-dimensional kagome ferromagnet Fe_3Ge ”, R. Lou et al. report on the effects spin reorientation has on the electronic structure in a three-dimensional (3D) kagome ferromagnet Fe_3Ge . The system is investigated with a combination of angle-resolved photoemission spectroscopy (ARPES) and density functional theory (DFT), with supporting characterization done by low-energy electron diffraction (LEED), and electrical resistivity measurements. Results are also compared to the known literature results on FeGe , a novel charge-density wave (CDW) kagome metal in order to explain absence of CDW in Fe_3Ge , proposing that van Hove singularities near the Fermi level play an important part in driving the CDW in Kagome systems.

While the presented results are generally interesting, the article is in my opinion more suitable for a more specialized journal. In particular, I believe it would be beneficial that authors presents at least some of their data on CDW in the main body of the article to go with their detailed discussion, and not only have it in the supplementary material, as some of the images are relevant for understanding the text. My further comments and suggestions can be found below:

Reply: We sincerely thank Reviewer #1 for the valuable comments and suggestions and the endorsement of the novelty of our work. Following Reviewer #1’s suggestion, we have moved the data and analysis on CDW to the main text of the revision, which have been provided as updated Fig. 5. We have also carefully revised our manuscript following the other suggestions and comments from Reviewer #1, please see our point-by-point response in the following.

1. Samples are prepared by polishing, and then series of sputtering and annealing cycles. Are the authors sure that there is no change in the stoichiometry following the procedure? It is not uncommon for some species to be sputtered away faster than others.

Figure R1 | a,b, XPS spectra of the Ge 2p (a) and Fe 2p (b) core levels measured on the mechanically cleaned and sputtered-annealed crystal surfaces of Fe_3Ge using an $\text{Al } K\alpha$ X-ray source, respectively.

Reply: We thank Reviewer #1 for raising this concern. To demonstrate the negligible effect of sputtering and annealing procedures on the sample stoichiometry, we have conducted the comparative x-ray photoelectron spectroscopy (XPS) measurements on the surfaces without and with the sputter-anneal cycles. First, we mechanically cleaned *in situ* the surface of an as-grown Fe_3Ge single crystal by means

of a diamond needle file, and recorded the XPS spectra on it; then, we processed this clean surface using the same sputter-anneal cycles as in the main text, and performed similar XPS measurements. As shown in Fig. R1, the overall line shapes from these two surfaces are quite similar for both the Ge-2*p* and Fe-2*p* core levels, implying a negligible change of the sample composition. Moreover, as presented in the last version, we do not observe any additional reconstructions in the LEED patterns and the ARPES spectra show a good overall agreement with the DFT calculations. Therefore, we believe that the stoichiometry of Fe₃Ge crystals is little affected by the sputter-anneal treatments.

We have added Fig. R1 into updated Fig. S2. The corresponding discussion has also been included.

2. Related to question 1, and presented XPS data in Figure S2. I am surprised by a large difference in binding energy of bulk and surface component, as well as the difference in their intensity. Are authors sure that these are indeed bulk and surface components, and not Ge in different chemical environment? Furthermore, it would be nice to see these components fit.

Figure R2 | Multipieak fitting of the Ge 3*d* core-level spectra by six Lorentzian peaks (shades) with a linear background (orange line). The asymmetry was introduced into the Lorentzian peaks to capture the tail on the high binding energy side.

Reply: Following Reviewer #1's suggestion, we have carried out the fit of Ge-3*d* core level spectra. As shown in Fig. R2, the spectra can be well reproduced by three spin-orbit split doublets with a linear background. The values of spin-orbit splitting (~ 0.55 eV) and branching ratio (~ 1.81) of each doublet are comparable to the typical values of Ge 3*d* states reported in other materials, like the Ge(111) [*Phys. Rev. B* **48**, 2012(R) (1993)]. In general, the XPS intensity ratio between bulk and surface components depends sensitively on the emission angle (the angle between the sample surface normal and the lens axis of analyzer); as the emission angle decreases, the bulk sensitivity of the measurements is enhanced, leading to the enhancement of the bulk-to-surface intensity ratio. In the present case, our XPS measurements were performed near the normal emission angle (0°), the observed difference in the intensity of bulk and surface components is therefore reasonable. Meanwhile, according to our fitting results, the energy difference (absolute value) between bulk and surface components is about 0.24 eV, also agreeing well with the reported values in Ge(111) [*Phys. Rev. B* **48**, 2012(R) (1993)]. Regarding the doublet with tiny intensities at lower binding energies (green shades in Fig. R2), we infer that it comes from (i) the defects during bulk crystal growth and/or (ii) the defects/adatoms on the surface introduced by the polishing and/or the sputter-anneal treatments. Nonetheless, it is noted that such defects made no noticeable contribution to our ARPES spectra.

We have added Fig. R2 into updated Fig. S2. The corresponding discussion has also been included.

3. Regarding the choice of photon energy: While 125 eV and 146 eV are indeed close to k_z values of 0 and π , is there a particular reason that measurements were done in vicinity of these planes, and not exactly at them?

Reply: We thank Reviewer #1 for this question. For ARPES, the k_z conservation is not strictly conserved, and thus cause k_z broadening. As a result, the ARPES spectra reflects the electronic states integrated over a certain k_z region of bulk Brillouin zone and the electronic states at $k_z = 0$ and π have main contributions. Namely, the respective band features from $k_z = 0$ and π planes can be prominently resolved by a certain range of photon energies around 0 and π . This effect is not ideal but also gives us advantage not having to be at the exact photon energy to do the measurement. Meanwhile, according to the free electron final-state model, the determination of k_z in ARPES is actually an estimate based on an empirical value of the inner potential. These facts thus ensure the validity of the photon energies we used.

4. Often, data at different photon energies is presented in the figures (matching approximately $\pi/2$ plane), but no particular reason is given for this (for example Figure 2). Can authors explain the choice?

Reply: We thank Reviewer #1 for pointing this out. As said in the last version, due to the k_z broadening effect, the spectra taken with 135-eV photons, which correspond to the k_z value of $\lesssim \pi/2$, show many similarities with that from the $k_z \sim \pi$ plane. We presented the ARPES spectra under 135-eV photons (Figs. 2e and S4d,e) for two reasons: one is to show the dispersion connecting the DP2 and vHS2, which is much clearer than that under 146-eV photons; and the other is, more importantly, to illustrate that the ARPES intensity in the $k_z \sim \pi$ plane could suffer from a stronger k_z broadening effect than the $k_z \sim 0$ plane, which is critical to understand why the Dirac gap is experimentally observable at DP1 while not at DP2. To clarify this point, we have updated the caption of Fig. 2e accordingly.

5. Can authors elaborate on how measurements with different photon polarization and photon energy show that flat band is intrinsic to the kagome plane?

Reply: As concerned by Reviewer #1, we realized that the kagome flat band can sometimes be invisible in ARPES measurements with certain photon polarization [like the case in FeSn, *Nat. Mater.* **19**, 163-169 (2020)] and/or photon energy [like the case in YMn₆Sn₆, *Nat. Commun.* **12**, 3129 (2021)] due to matrix element effects. Therefore, to avoid the ambiguity, we have removed the sentence “further showing that the FB is intrinsic to the kagome lattice”.

6. Am I correct to assume that the “nearly gapless nature of DP1 (0.5 meV)” is determined from the DFT calculations? The experimental energy resolution is 5 meV, and this is for sure too small gap to be resolved.

Reply: Yes, the Dirac gap (~ 0.5 meV) at DP1 in the planar ferromagnetic state is determined from the DFT calculations. We have modified the corresponding sentence to “compatible with the nearly gapless nature of the DP1 in the planar FM state expected from the DFT calculations (~ 0.5 meV, Fig. 1e,f)”.

7. Page 10, regarding canting of the Fe moments: Authors state that change of band gap at the DP1 indicates that Fe moments start to cant away from the xy plane towards the z axis around 170 K. Is this correct? Based on the Figure 1, Fe moments are already canted at 380 K, and below 170 K they should be in the xy plane, and not canting towards z axis.

Reply: We thank Reviewer #1 for raising this concern. In the last version, our point is that, upon warming up, once the temperature is above ~ 170 K, the Fe moments start to cant away from the xy plane towards the z axis. This is consistent with the magnetic phase diagram sketched in Fig. 1c. To avoid the ambiguity, we have modified the corresponding sentence to “This remarkable change of DP1 is in line with the theoretical expectation (Fig. 1e-g) that the Dirac gap at DP1 can be tuned by the spin reorientation; accordingly, it can also be obtained that, once the temperature is increased above ~ 170 K, the Fe moments start to cant away from the xy plane towards the z axis.”

8. While k_z broadening is a serious issue when it comes to ARPES measurements of 3D materials with VUV light, I am wondering if authors have an explanation why the broadening seems to be significant around $k_z = \pi$ and not around $k_z = 0$?

Reply: We appreciate Reviewer #1 for this comment. As discussed in our response to the 4th question of Reviewer #1, the observations under 135-eV photons not only demonstrate the existence of k_z broadening, but additionally suggest that the projected electronic states from $k_z = \pi$ plane make a dominant contribution over a wide k_z interval which is larger than $\pi \pm \pi/2$, namely, the ARPES intensity in $k_z \sim \pi$ suffers from a stronger k_z broadening effect than $k_z \sim 0$.

The distinct strength of k_z broadening around 0 and π is due to the overall bands along k_z direction dispersing steeply around π while becoming relatively gently dispersive around 0, as seen from the DFT band structures along Γ -A direction in Fig. 1e-g. As a result, the k_z broadening effect becomes more visible around $k_z \sim \pi$ plane than around $k_z \sim 0$ plane. In the revised manuscript, we have added more discussion on the k_z broadening effect to clarify this point.

9. Discussion of the CDW states heavily relies on the data provided in the supplementary information, and is not so easy to follow without it.

Reply: We thank Reviewer #1 for pointing this out. Following Reviewer #1's advice, in the revised manuscript, we have moved the data and analysis on CDW to the main text, which have been provided as updated Fig. 5. Further, we have amended the manuscript (article title, abstract, text) accordingly, e.g., the title has been modified to “Orbital-selective effect of spin reorientation on the Dirac fermions in a non-charge-ordered kagome ferromagnet Fe_3Ge ”.

10. Figure 1: I am not sure that I am able to see any difference between figures e) and f). I would recommend that authors either highlight some differences, or at least clarify in the text whether there are any. Furthermore, while the inset zooming in on the DP1 is helpful in seeing that there is crossing, no gap is visible in it (and I assume this is where we expect DP1 to have 0.5 meV gap).

Reply: Yes, there is no noticeable difference between Figs. 1e and 1f, indicating that, in the planar

ferromagnetic state of Fe₃Ge, the orientation of Fe moments has a negligible effect on the overall electronic structure. Following Reviewer #1's suggestions, we have clarified this point in the revision; and we have further zoomed in on the DP1 in the insets of Fig. 1e,f, the tiny Dirac gap can be clearly revealed now.

11. Figure 2: Is there a particular reason that k-range in subfigures f) and g) is different? It seems that the only difference is that data in f) is measured with vertically polarized light and in g) with horizontally polarized light, so the k-range should also be the same for the two.

Reply: We thank Reviewer #1 for this question. As indicated in Fig. 2f,g, these two spectra were actually measured along different high-symmetry directions, with Fig. 2f along the Γ -K-M line and Fig. 2g along the Γ -M line.

12. Figure 2: While it is possible to see what bands are used to fit the parabola at $k_x = k_v - 0.25\text{\AA}^{-1}$, it is not possible to see what band is used at $k_x = k_v - 0.13\text{\AA}^{-1}$. Also, what is the nature of the more intense, hole-like band visible at these cuts? Finally, it is not clear to me what k_v actually stands for.

Reply: We appreciate Reviewer #1 for raising these concerns. To help readers better resolve the hole bands associated with the vHS1, we now only present half of the parabolic curve in each panel of updated Fig. 2h. We have also updated the caption of Fig. 2h to clarify that k_v stands for the momentum value (k_x direction) of vHS1. As for the intense hole-like band concerned by Reviewer #1, it is a part of the dispersion that forms the inner hole-like Fermi surface contour around Γ point. As indicated in Fig. R3a, we measured several cuts parallel to the K-M-K direction further away from the vHS1. In Fig. R3b, one can see that, upon approaching Γ point, the concerned hole-like band gradually disperses upwards to cross E_F , eventually forming the inner hole-like Fermi pocket around Γ point.

Figure R3 | a, Fermi surface mapping of Fe₃Ge taken by the 125-eV photons with linear horizontal

polarization, adopted from Fig. 2a. **b**, ARPES intensity plots measured along cuts 1-4 in **a**, respectively.

13. Figure 3: I am slightly confused about the image in 3f). It has the same oval shape that was used in 3a) to indicate where the MDC is taken. Is this indeed along Γ -M- Γ -M direction, or the direction it is indicated in 3a)? Is it is former, it would be nice to also see ARPES intensity map in this direction.

Reply: Yes, the MDCs in Fig. 3f are taken along the Γ -M- Γ -M direction. In the last version, this momentum direction was indicated by the capsule-like marker together with the black dashed line at E_F in Fig. 3a. To avoid the possible ambiguity, in the revised manuscript, we have replaced the black dashed line at E_F (Fig. 3a) with a cyan dashed line and added a cyan dashed line also onto Fig. 3f; we have added the capsule-like marker also onto Fig. 2g to further indicate this direction; the corresponding text and the captions of Fig. 3a,f have also been modified.

14. Figure 4: It might be more informative to have larger images of orbital nature around the points of interest (DP1, DP2 vHS etc), as it is not possible to see some of the orbital contributions without zooming in quite a lot the image.

Reply: We appreciate Reviewer #1 for this comment. The electronic structures with spectral weight projected onto five Fe 3d orbitals have now been plotted separately in updated Figs. 4 (M_y) and S9 (M_x).

15. Figure 4: Regarding orbital contribution for vHS1, it seems like it has contribution not only from Fe-3d_{xy} and Fe-3d_{x²-y²}, but also Fe-3d_{z²} orbitals. Again, even when zooming in a lot, it is hard to really see what contributes most.

Reply: We thank Reviewer #1 again for this comment. According to the updated orbital-projected DFT calculations, the vHS1 is indeed contributed by the Fe-3d_{xy}, Fe-3d_{x²-y²}, and Fe-3d_{z²} orbitals. More importantly, the orbital differentiation between the DP1 and DP2 is actually derived from their different out-of-plane orbital characters, with the DP1 mainly of the Fe-3d_{xy}, Fe-3d_{x²-y²}, and Fe-3d_{z²} orbitals and the DP2 mainly of the Fe-3d_{xy}, Fe-3d_{x²-y²}, Fe-3d_{xz}, and Fe-3d_{yz} orbitals. We apologize for the incorrect orbital information delivered by the last version, which is imputed to the previous data presentation of the orbital-resolved calculations. The contributions of different orbitals overlaid each other therein, thus interfering with our previous assignments. In the revised manuscript, we have corrected all orbital characters and modified the corresponding discussions properly.

Reviewer #2 (Remarks to the Author):

Materials containing kagome lattice have attracted significant scientific interest in recent years owing to their unique electronic structures, topological properties, magnetism and superconductivity. In this manuscript, the authors studied the band structure of a kagome ferromagnet Fe3Ge with angle-resolved photoemission spectroscopy (ARPES) and density functional theory calculations. Due to the change of temperature-induced magnetic moment direction in this system, they can capture the effect of spin-reorientation on the band structure through the ARPES experiments. The main result of the article is to track the band structures on different k_z planes, indicating that spin reorientation on Dirac fermions has orbital selectivity. However, Dirac points are not rare and appear in various systems, including honeycomb,

triangular and kagome lattices, etc. In addition, the strong three-dimensional nature of Fe₃Ge makes the band structure very complex and the kagome band characteristics are not obvious compared with its sibling compound FeGe. The Dirac points studied in this manuscript are difficult to identify both theoretically and experimentally. I believe that this manuscript lacks significant findings and sufficient novelty to be published in Nature Communications. The detailed concerns and reasons are listed as follows:

Reply: We sincerely thank Reviewer #2 for the careful review and the valuable comments and suggestions of our manuscript. First and foremost, it is imperative to emphasize that our work provides profound insights into not only the manipulating of typical kagome electronic structure and its topological characters but additionally the understanding of novel charge order in kagome magnet FeGe. As Reviewer #3 also pointed out: "*the results presented here represent an intriguing advancement in the understanding of the physics of these relatively novel class of correlated materials*". In our opinion, this is mainly because the realization of kagome materials possessing tunable kagome bands and topological properties via the clean tuning knobs, like the magnetism, is exceptionally rare, and any close sibling of FeGe has not been established thus far, which hinders the elucidation of the origin of charge order therein. Therefore, the two main findings of our work offer important information to understand the topological physics and the CDW instabilities in magnetic kagome lattices.

We agree with Reviewer #2 that the observation of Dirac point is not rare in kagome lattices. However, up to the present time, the tuning of Dirac gap by the intrinsic magnetism has yet to be experimentally realized within a single kagome system. Such experimental realization constitutes one of the central focuses of our research. Moreover, despite the three-dimensional nature of Fe₃Ge as raised by Reviewer #2, the overall band structure is found to be relatively simple near E_F , especially true for the $k_z \sim 0$ plane where the temperature evolution of Dirac gap (DP1) is revealed. This ensures a reasonable general agreement between experimental observations and DFT calculations, in particular the kagome band characteristics. As a result, two sets of Dirac points and van Hove singularities (DP1,2 and vHS1,2) together with the kagome flat band have been unambiguously identified in our work.

To also emphasize the importance of our finding on the CDW order, in the revised manuscript, we have moved the data and analysis on CDW to the main text, which have been provided as updated Fig. 5; further, we have amended the manuscript (article title, abstract, text) accordingly, e.g., the title has been modified to "Orbital-selective effect of spin reorientation on the Dirac fermions in a non-charge-ordered kagome ferromagnet Fe₃Ge".

1. The authors claim that "The frustrated lattice geometry of a kagome lattice can give rise to the unique electronic structure embracing the flat band..." on line 6 of page 4 in the main text. It is misleading because the frustrated lattice geometry can bring the frustrated magnetism, but is not the reason for the kagome band structure.

Reply: We appreciate Reviewer #2 for this comment. We have modified the corresponding sentences to "A tight-binding model on the kagome lattice yields a symmetry-protected electronic structure hosting the flat band (FB) over the entire Brillouin zone (BZ), the Dirac point (DP) at the BZ corner, and the van Hove singularities (vHSs) at the BZ boundary. A further combination of its unique lattice geometry and the intrinsic magnetism has been reported to engender a variety of novel quantum phenomena, ...".

2. The last sentence on page 8 states that the flat band near DP1 and vHS1 is not attributed to destructive interference. The author should clarify the underlying reasons for this observation.

Reply: We thank Reviewer #2 for this comment. As discussed in the previous sentence of the concerned one, due to the small difference in the binding energies of DP1 and vHS1 as well as the DP1 consisting of less-dispersive bands as suggested by the DFT calculations, the dispersion that connects the DP1 and vHS1 is nearly flat and almost indistinguishable from the lower branch of DP1 along the *K-M* direction. Consequently, these two dispersions nearly overlap with each other, exhibiting as the flat spectral feature between *K* and *M* points.

To avoid the ambiguity, we have removed the concerned sentence and modified its previous sentence according to the above discussion.

3. How to explain the small lift of band degeneracy in paramagnetic state along L-H-A paths in Fig. S3a? This does not appear in the ferromagnetic bands in Figs. 1e-f.

Reply: We thank Reviewer #2 for this question. The magnetic configurations of Fe₃Ge with different directions lead to different magnetic space groups (MSGs), making the little groups on certain high-symmetry paths with different symmetries. As shown in Tables R1-R3, since the bands of Fe₃Ge with the ferromagnetic (FM) moments along *x* ([100]), *y* ([010]) and *z* ([001]) directions or in paramagnetic (PM) phase have different little groups along the Γ -A, L-H, and H-A paths, they have different degeneracies on different paths.

To clarify this point, we have added Tables R1-R3 and corresponding discussion into the updated Supplementary Information.

Γ -A (0, 0, w)

Phase	MSG	Unitary subgroup	Magnetic little co-group	Little co-group	Little group
PM	P6 ₃ /mmc1' (194.264)	194	6/m'mm	6mm	C6v
FM[100]	Cmc'm' (63.463)	12	m'm2'	m	Cs
FM[010]	Cm'cm' (63.464)	15	m'm2'	m	Cs
FM[001]	P6 ₃ /mm'c' (194.270)	176	62'2'	6	C6

L-H (u, v, 1/2)

Phase	MSG	Unitary subgroup	Magnetic little co-group	Little co-group	Little group
PM	P6 ₃ /mmc1' (194.264)	194	2'/m	m	C2v
FM[100]	Cmc'm' (63.463)	12	2'	1	C1
FM[010]	Cm'cm' (63.464)	15	2'	1	C1
FM[001]	P6 ₃ /mm'c' (194.270)	176	m	m	Cs

H-A (u, u, 1/2)

Phase	MSG	Unitary subgroup	Magnetic little co-group	Little co-group	Little group
PM	$P6_3/mmc1'$ (194.264)	194	m'mm	mm2	C2v
FM[100]	Cmc'm' (63.463)	12	/	/	C1
FM[010]	Cm'cm' (63.464)	15	/	/	C1
FM[001]	$P6_3/mm'c'$ (194.270)	176	m'm2'	m	Cs

Table R1-R3 | MSGs, unitary subgroups, magnetic little co-groups, little co-groups, and little groups of different magnetic configurations along the Γ -A (**Table R1**), *L-H* (**Table R2**) and *H-A* (**Table R3**) paths, respectively.

4. The Fermi surfaces on the $k_z = \pi$ plane in Fig. 2b and Fig. S4 show very rich spectral patterns, which are not consistent with calculated band structures in Fig. 1. For example, three Fermi surface pockets cross the A-H path appear in Fig. S4b, while only two bands intersect with the Fermi level in Figs. 1e and 1f. It is necessary to calculate the Fermi surface on the $k_z = \pi$ plane and compare it with the experimental results.

Reply: We appreciate Reviewer #2 for this comment. Following Reviewer #2's suggestion, we have calculated the bulk Fermi surface of Fe_3Ge in the $k_z = \pi$ plane, as shown in Fig. R4. By comparing the ARPES mappings and DFT Fermi surfaces, one finds the electron-like pocket around *H* point, which is derived from the Dirac bands of DP2, matches well with the DFT calculations with a slight difference in the pocket size, whose origin is discussed in our response to the 5th question of Reviewer #2. On the other hand, since the ARPES intensity in $k_z \sim \pi$ suffers from a stronger k_z broadening effect than $k_z \sim 0$ (please refer to our response to the 8th question of Reviewer #1 for details), it is thus inferred that the other rich topologies in Figs. 2b and S4b-e, which are not reproduced in Fig. R4, could be the projections from other k_z planes.

To clarify this point, in the revised manuscript, we have added Fig. R4 into updated Fig. S5 and included more discussion on the Fermi surfaces in the $k_z \sim \pi$ plane.

Figure R4 | **a**, DFT calculated bulk Fermi surfaces of Fe_3Ge in the $k_z = \pi$ plane. The calculations were carried out by considering a FM moment ($\mu_{Fe} \approx 2.2\mu_B$) aligned along the *x* axis. **b**, Same as **a** with the FM moment along the *y* axis.

5. The energy gap at DP2 in the calculated band structure was not observed in the experiment, which is attributed to the k_z broadening effect. However, DP2 appears at different energies of about $E - E_F = -0.1\text{eV}$ and -0.5eV in the calculated and experimental band structures, respectively. This obvious discrepancy should be explained.

Reply: We appreciate Reviewer #2 for this comment. First, we would like to point out that the assignment of DP2 in experiment is well substantiated for the following reasons. (i) The connectivity between DP2 and vHS2 in DFT (Fig. 1e,f) is clearly revealed by ARPES (Fig. 2d,e), this is in line with the tight-binding band structure of kagome lattice. Moreover, the near- E_F hole-like band (δ) around L point dispersing towards the lower branch of DP2 (Fig. 1e,f) is also clearly captured by ARPES (Fig. 2d,e). (ii) According to the DFT calculations (Fig. 1e,f), the upper branch of DP2 hosts different Fermi velocities (v_F 's) along the $L-H$ and $H-A$ directions, with the v_F for the former is slightly larger than the latter (updated Table S4). Again, this property is unambiguously observed by our ARPES measurements (Fig. 3h and updated Table S4).

Regarding the difference between experiment and calculation in the binding energy of DP2 as concerned by Reviewer #2, it is most likely due to the presence of moderate electron-electron correlations in Fe_3Ge which are not included in the DFT calculations. With a bandwidth renormalization factor of about 3 (updated Table S4), the overall ARPES spectra could show a markedly reduced bandwidth compared with that in the DFT calculations, thus giving rise to the mentioned discrepancy. In the revised manuscript, we have added the corresponding discussion to clarify this point.

6. It seems that only one energy band passes through the DP1 in the experimental band dispersion of Fig. 3a. If there is no separated bands near DP1, how can we experimentally confirm that it is a Dirac point?

Figure R5 | Energy distribution curves of the ARPES spectra taken along the K - M direction. The black solid circles indicate the band dispersions associated with the DP1. The separation can hardly be resolved in the momentum range from $\sim 1/3MK$ to K point.

Reply: We thank Reviewer #2 for this comment. As discussed in our response to the 2nd question of Reviewer #2, along the K - M direction, the two dispersions forming the DP1 are nearly flat and almost overlap with each other. But the separation between these two bands can still be resolved when approaching M point, with one forming the vHS1 and the other merging with the β band (Fig. 3a). Such separation near M point is further demonstrated by the corresponding energy distribution curves presented

in Fig. R5. As for the Γ - K direction, one branch of the DP1 is experimentally revealed (γ band) while the other branch is not visible in ARPES due to matrix element effects.

In the revised manuscript, we have provided Fig. R5 as updated Fig. S6 and added more discussion on the DP1.

7. The authors claim that DP1 is mainly contributed by a combination of the Fe- d_{xy} and Fe- $d_{x^2-y^2}$ orbitals, while DP2 is primarily associated with the Fe- $d_{x^2-y^2}$ orbital. However, in Fig. 4a, the Fe- d_{xy} orbital colored with red dots also seems to dramatically contribute to the DP2. This directly affects the main conclusion of this manuscript, which is that the effect of spin reorientation on Dirac fermions has orbital selectivity. To clarify this point, electronic band structures weighted by projected different Fe orbitals should be plotted separately.

Reply: We thank Reviewer #2 for this comment. Following Reviewer #2's suggestion, the electronic structures with spectral weight projected onto five Fe 3d orbitals have now been plotted separately in updated Figs. 4 (M_y) and S9 (M_x). According to the updated orbital-projected DFT calculations, the orbital differentiation between the DP1 and DP2 is actually derived from their different out-of-plane orbital characters, with the DP1 mainly of the Fe- $3d_{xy}$, Fe- $3d_{x^2-y^2}$, and Fe- $3d_{z^2}$ orbitals and the DP2 mainly of the Fe- $3d_{xy}$, Fe- $3d_{x^2-y^2}$, Fe- $3d_{xz}$, and Fe- $3d_{yz}$ orbitals.

As a result, our conclusion about the orbital-selective effect of spin reorientation on Dirac fermions has not been affected. We apologize for the incorrect orbital information delivered by the last version, which is imputed to the previous data presentation of the orbital-resolved calculations. The contributions of different orbitals overlaid each other therein, thus interfering with our previous assignments. In the revised manuscript, we have corrected all orbital characters and modified the corresponding discussions properly.

Reviewer #3 (Remarks to the Author):

This work focuses on the angle-resolved-photoemission study of the Kagome ferromagnet Fe₃Ge, reporting an orbital-selective effect of the temperature-dependent spin reorientation on the Dirac fermions. The data presented in the Main text and Supplementary are of high quality and comprehensive, supporting the Authors' main conclusions. I believe the results presented here represent an intriguing advancement in the understanding of the physics of these relatively novel class of correlated materials; therefore, I recommend the manuscript for publication in Nature Communications.

Reply: We sincerely thank Reviewer #3 for the careful review and the positive evaluation of our results. We have carefully revised the manuscript following their valuable comments and suggestions.

However, I do have few comments I would like the Authors' to address to strengthen the stated conclusions:

- One of the main results is the identification of a two-peak structure in the EDC of DP1 at 210K, presented in Fig.3c and S7. However, it is based only on one temperature data point (210K). Are there additional measurements taken at similar or higher temperatures which can confirm the trend?

Reply: We appreciate Reviewer #3 for this comment. We have carried out similar ARPES measurements

along the Γ - K - M direction at $T = 220$ K. As shown in Fig. R6a, the line shape of the energy distribution curve (EDC) taken at K point is similar to that of 210 K (Fig. 3c and updated Fig. S8a), in particular the double-hump structure at DP1. In Fig. R6b, we fit the data near E_F in the same fashion as before. The obtained DP1 gap size (~ 50 meV) is comparable to the gap value at 210 K, confirming its trend with the spin reorientation. To clarify this point, we have added Fig. R6 into updated Fig. S8 and included corresponding discussion in the updated Supplementary Information.

Figure R6 | a, EDC taken at K point with the corresponding ARPES spectra recorded along the Γ - K - M direction ($h\nu = 125$ eV, LH polarization, $T = 220$ K). The black dashes are extracted peak positions. **b**, Quantitative fitting of the data in **a** by two Gaussian peaks (grey solid curves) and a background (yellow solid curve), which is modeled by considering a polynomial function together with the Fermi-Dirac distribution. The fitting result is superimposed as the black solid curve. An energy gap of ~ 50 meV can be obtained at the DP1.

- On the same topic, Fig. S7 shows the result of the EDC fit with two Gaussians, along the individual components. I wonder what a similar fit with a single Gaussian component would look like, and how it would compare to the 2-peaks one in terms of Chi-squares or residuals. I think an argument like this would greatly reinforce the Authors' statement.

Reply: We appreciate Reviewer #3 for this suggestion. As presented in Fig. R7b, we have carried out the fit of EDCs (210 K) by using one Gaussian peak. In Fig. R7c,d, we plot the raw data together with the two-peak and one-peak fitting results. One can see obvious deviations between one-peak fit and raw data in the energy range from ~ -0.1 to ~ -0.2 eV, in sharp contrast to the overall goodness of the two-peak fit. This is clearly reflected in the difference curves (bottom dashed curves), where the absolute values of differentiation for the one-peak fit are generally larger than the two-peak one. To quantify the goodness of these two fits, we calculated the chi-square (χ^2) values as suggested by Reviewer #3. As shown in Fig. R7a,b, the χ^2 values of the two-peak fit are much smaller than that of the one-peak fit, further validating our assessments. To clarify this point, we have added Fig. R7 into updated Fig. S8 and included corresponding discussion in the updated Supplementary Information.

Figure R7 | a,b, Quantitative fitting of the EDCs (210 K) at K point by using two Gaussian peaks (a) and one Gaussian peak (b), respectively. The background (yellow solid curves) is modeled by considering a polynomial function together with the Fermi-Dirac distribution. The fitting results are superimposed as the black solid curves. **c,d,** EDCs (210 K) from updated Fig. S8a (c) and Fig. 3c (d), respectively, together with the corresponding two-peak and one-peak fits. The dashed curves represent the difference between raw data and fitting results.

• The Authors' attribute the lack of DP2 gap in the experiment to k_z projection broadening. How could be excluded that the same effect is taking place at DP1, affecting the probed dispersion?

Reply: We thank Reviewer #3 for this comment. As discussed in the main text and our response to the 8th question of Reviewer #1, we find that the ARPES intensity in $k_z \sim \pi$ suffers from a stronger k_z broadening effect than $k_z \sim 0$, this is reflected in the following two aspects. (i) The projected electronic states from $k_z = \pi$ plane make a dominant contribution over a wide k_z interval which is larger than $\pi \pm \pi/2$. (ii) The ARPES mapping and band structure in the $k_z \sim 0$ plane (Figs. 2a and 3a) are relatively simple and in good agreement with the DFT calculations (Figs. 2c and 3b), while the ARPES data in the $k_z \sim \pi$ plane (Figs. 2b and S4b,c) show very rich spectral features and many of them are not well reproduced by DFT (updated Fig. S5c,d). The distinct strength of k_z broadening around 0 and π is due to the overall bands along k_z direction dispersing steeply around π while becoming relatively gently dispersive around 0, as seen from the DFT band structures along Γ -A direction in Fig. 1e-g. Therefore, the ARPES spectra in $k_z \sim$

0 are less affected by the k_z broadening, ensuring that our observation of the temperature-dependent gap at the DP1 is intrinsic. To clarify this point, in the revised manuscript, we have added more discussion on the k_z broadening effect.

List of changes to manuscript

- We added or revised the following figures and tables:
 1. Insets of Figs. 1e and 1f to show the tiny Dirac gap at DP1 in the planar FM state;
 2. Fig. 2a to clarify what k_v stands for;
 3. Fig. 2c to clarify the k_z position of the calculated Fermi surface;
 4. Fig. 2g to show the location of the extracted MDCs in Fig. 3f;
 5. Fig. 2h to better resolve the hole bands associated with the vHS1;
 6. Fig. 3a,f to show the momentum direction of the temperature-dependent MDCs;
 7. Fig. 3h to illustrate the estimate of the location of vHS3 slightly above E_F ;
 8. Fig. 4a-e to add the electronic structures with spectral weight projected onto five Fe 3d orbitals separately (M_y);
 9. Fig. 5a,b to add the autocorrelation of the ARPES intensity maps;
 10. Fig. 5c-f to add the temperature-dependent EDCs and symmetrized EDCs taken at k_F 's of the vHS1 and vHS3 bands;
 11. Fig. 5g,h to add the temperature dependence of the zero-field electrical resistivity;
 12. Fig. S2a,b to add the evidence for the little effect of sputter-anneal treatments on the sample stoichiometry;
 13. Fig. S2d to add the quantitative fitting of the Ge 3d core-level spectra;
 14. Fig. S5a,b to clarify the k_z position of the calculated Fermi surfaces;
 15. Fig. S5c,d to add the DFT calculated Fermi surfaces in the $k_z = \pi$ plane;
 16. Fig. S6 to add the evidence for the two energy bands passing through the DP1;
 17. Fig. S8c to add the quantitative fitting of the EDCs at DP1 with one Gaussian peak;
 18. Fig. S8b,c to show the chi-square value of each fit;
 19. Fig. S8d,e to add the detailed comparison between the two-peak and one-peak fits;
 20. Fig. S8f,g to add the evidence for the similar DP1 gap opening at higher temperature;
 21. Fig. S9a-e to add the electronic structures with spectral weight projected onto five Fe 3d orbitals separately (M_x);
 22. Fig. S10a-d to show the DFT band structure calculations with the Fe-3d and Ge-4p orbital projections;
 23. Fig. S12 to add the DFT calculations without the spin-orbit coupling effect;
 24. Table S1-S3 to add the representation analysis of different magnetic configurations.
- We added the following sections into the Supplementary Information:
 1. Note 1 - Stable stoichiometry and detailed analysis of the X-ray photoelectron spectroscopy data;
 2. Note 2 - Different band degeneracies between different magnetic states;
 3. Note 3 - Reproducibility, temperature evolution, and quantitative analysis of the DP1 gap.
- Major revisions to the text of the main manuscript and the Supplementary Information (used a red font to mark out).

Reviewer #1 (Remarks to the Author):

The resubmitted draft by R. Lou et al. of the "Orbital-selective effect of spin reorientation on the Dirac fermions in a non-charge-ordered kagome ferromagnet Fe₃Ge" answers majority of my questions (not all), and before I can recommend publication in Nature Communications, I would like to have remaining questions answered and clarified.

Reply: We sincerely thank Reviewer #1 for the careful review and the positive recommendation for the publication of our manuscript after revision. We have carefully revised the manuscript following their valuable comments and suggestions.

1. Abstract states "Moreover, we present detailed comparative studies between the non-charge-ordered Fe₃Ge and its sibling compound FeGe, a newly established charge-density-wave kagome magnet, our results strongly support that the orbital-selective van Hove singularities near the Fermi level play an indispensable part in driving the novel charge order on a magnetic kagome lattice." I find this slightly misleading and would suggest rephrasing it. The presented work is done on solely Fe₃Ge, and FeGe comparison is fairly brief. Furthermore, comparison is done in comparison with various publications on FeGe, and no FeGe data is presented here.

Reply: We appreciate Reviewer #1 for this comment. We have modified the concerned sentences to "Moreover, we demonstrate that there is no signature of charge order formation in Fe₃Ge, contrasting with its sibling compound FeGe, a newly established charge-density-wave kagome magnet. Our detailed studies of Fe₃Ge in this regard support that the orbital-selective van Hove singularities near the Fermi level play an indispensable part in driving the novel charge order on a magnetic kagome lattice." We have also properly modified the corresponding sentences in the introduction, discussion, and conclusion.

2. Page 10: As for the Γ -K direction, one branch of the DP1 is experimentally revealed (ϵ band, Fig. 3a) while the other branch is not visible in ARPES due to matrix element effects.

Authors state that they did polarization dependent measurements, and a fair amount of data in LV polarisation is presented in the supplementary information. I would like to know if they see the second band in LV polarisation, as this is not mentioned in either the main text nor in the supplementary information? If they do see the band this should be shown at least in the supplementary data. If they do not see it with LV, reasons for this need to be clarified, as so far it is only stated that band is not visible with LH due to matrix polarisation effects, which leads one to believe it should be visible with LV.

Reply: We thank Reviewer #1 for pointing out this ambiguity. In the last version, our point is actually that, due to matrix element effects, the concerned second branch (denoted as ϵ band) of DP1 is not visible along the Γ -K direction of the first Brillouin zone (BZ). This is justified by the Fermi surface topologies in experiments (Fig. 2a). More specifically, according to the DFT calculated Fermi surfaces (Fig. 2c) and band structures (Fig. 3b) in the $k_z = 0$ plane, one obtains that the two branches of DP1 along the Γ -K direction contribute to the two hole-like Fermi pockets around Γ point. As said in the main text (page 8, paragraph 2), these two Fermi pockets are clearly observed in our experiments (Fig. 2a); moreover, the outer pocket near the Γ -K direction, which corresponds to the ϵ band, is vanishingly weak in the first BZ while is more visible in the second BZ due to matrix element effects.

To validate the presence of ϵ band along the Γ - K direction, we have now measured the ARPES spectra in the second BZ. As shown in Fig. R1a,b, the two band dispersions (γ and ϵ , indicated by the red arrows) forming the DP1 along the $\Gamma(\Gamma')$ - K direction are clearly observed in the second BZ, consistent with the experimental Fermi surface topologies in Fig. 2a.

To avoid the ambiguity, we have provided Fig. R1 as updated Fig. S7 and included the corresponding discussion in the updated Supplementary Information; further, we have modified the concerned sentence to “As for the Γ - K direction, one branch of the DP1 is experimentally revealed (γ band, Fig. 3a), the other branch is not visible in the first BZ but in the second BZ due to matrix element effects (see Supplementary Fig. S7 and Note 3 for its presence in the second BZ).”

Figure R1 | a,b, ARPES intensity plot (a) and corresponding second derivative plot (b) recorded along the Γ' - K - M direction with the 125-eV photons (LH polarization), respectively. Γ' denotes the center of the second BZ. The red arrows indicate the two bands (γ and ϵ) forming the DP1 along the Γ' - K direction. The red curves are guides to the eye for the vHS1 band along the K - M - K direction and its connecting to the DP1.

3. Regarding k_z smearing: Caption and data for figure 2, and Supplementary information figure S4. When looking at S4 data for 116 eV vs 135 eV, one can see several differences in the electronic structure. Can the authors explain what is the reason that measurements were not performed at 116 eV, but were performed at 135 eV? I would really like to know explanation for this, as Figure 3h, 16 K measurements whose data is furthermore taken at 146 eV (which is actually at $k_z = \pi$, and not 135 eV (or 116 eV)).

Reply: We thank Reviewer #1 for raising this concern. In our manuscript, the 116-eV data are presented in Figs. 2d, S4b,c, and S8e,f, the 135-eV data are presented in Figs. 2e and S4d,e. For clarity, here we state the reason of each ARPES measurement performed at these two photon energies. As discussed in the last round of review, Figs. 2d and 2e are to show the dispersion connecting the DP2 and vHS2 more clearly, Fig. 2e is also to illustrate that the ARPES intensity in $k_z \sim \pi$ suffers from a stronger k_z broadening effect than $k_z \sim 0$. Figs. S4b,c and S4d,e are to show the circular electron-like pocket around H point, serving as a supplement to the Fermi surface data at 146 eV (Fig. 2b). Fig. S8e,f are to show the observation of the kagome flat band.

As mentioned by Reviewer #1, the k_z -projected spectra in the 116 eV-mappings (Fig. S4b,c) are generally more intense than that in the 135-eV (Fig. S4d,e) as well as the 146-eV mappings (Fig. 2b). Therefore, to

avoid complex band structures, instead of 116 eV, we used 146 eV for the high-resolution measurements and temperature-dependent measurements (Figs. 3 and 5) to quantitatively study the properties of DP2 and vHS3 in the $k_z \sim \pi$ plane. Although the spectra taken with 135-eV photons show many similarities with that from the $k_z \sim \pi$ plane, for the sake of rigour and clarity, we did not use 135 eV for these quantitative studies in the $k_z \sim \pi$ plane (Figs. 3 and 5) because it corresponds to the k_z value of $\lesssim \pi/2$.

4. Supplementary figure S6. I am not sure that I clearly see that there are 2 peaks close to the M point. Was this obtained via fitting or?

Reply: We appreciate Reviewer #1 for this comment. For clarity, as shown in Fig. R2b-e, we have now presented the quantitative fitting of the three individual energy distribution curves (EDCs) closest to M point (denoted as #2-#4 in Fig. R2a) as well as the individual EDC at K point (denoted as #1 in Fig. R2a). In Fig. R2b, the raw EDC (red markers) at K point is well fitted by one Gaussian peak (grey solid curve) with a background (yellow solid curve), which is modeled by considering a polynomial function together with the Fermi-Dirac distribution. The full width at half maximum (FWHM) of the Gaussian peak is of ~ 0.09 eV. In Fig. R2c-e, we fit the EDCs close to M point in the same fashion as Fig. R2b but with two Gaussian peaks. Note that, to ensure the validity of using two peaks in these fits, the FWHM of each Gaussian peak in Fig. R2c-e is constrained to the same value (~ 0.09 eV) as that in Fig. R2b. The obtained good fits (black solid curves) can validate our identification of the double-hump spectral feature near M point. As a result, the concerned separation when approaching M point is now well supported.

To clarify this point, in the revised manuscript, we have added Fig. R2b-e into updated Fig. S6.

Figure R2 | a, EDC plot of the ARPES spectra taken along the K - M direction. The black solid circles indicate the band dispersions associated with the DP1. The figure is adopted from previous Fig. S6. **b-e**, Quantitative fitting of the individual EDCs (#1-#4 in **a**) by using one Gaussian peak (#1, **b**) and two Gaussian peaks (#2-#4, **c-e**), respectively. The Gaussian peaks are displayed as the grey solid curves. The background (yellow solid curves) is modeled by considering a polynomial function together with the Fermi-Dirac distribution. The fitting results are superimposed as the black solid curves. To ensure the validity of using two peaks in the fits of the EDCs close to M point, the FWHM of each Gaussian peak in **c-e** is constrained to the same value (~ 0.09 eV) as that in **b**.

Reviewer #2 (Remarks to the Author):

I appreciate author's revisions to the manuscript. However, the revised manuscript still lacks enough novelties and interesting results. Compared to numerous systems featuring Dirac points, the current study on kagome Fe₃Ge does not appear particularly compelling. The 3D structural nature, complex band structure, and the deep energy levels of Dirac points of Fe₃Ge hinder further manipulation of these Dirac points and the observation of related quantum effects. Additionally, several other issues in the manuscript are listed below. With revisions, the manuscript could be suitable for submission to a specialized journal.

Reply: We sincerely thank Reviewer #2 for the careful review and the valuable comments and suggestions of our revised manuscript. However, we disagree respectfully with Reviewer #2's comment that our work "lacks enough novelties and interesting results". We would like to point out that the novelty of our manuscript has been recognized by Reviewers #1 and #3, as Reviewer #1 stated "***the presented results are generally interesting***" and Reviewer #3 stated "***I believe the results presented here represent an intriguing advancement in the understanding of the physics of these relatively novel class of correlated materials***".

As discussed in the last round of review, going beyond the observation of kagome electronic structure reported in previous studies, our work for the first time demonstrates the tunability of the kagome bands and topological characters via the internal magnetism in a single kagome material.

(i) Despite the three-dimensional (3D) nature of Fe₃Ge, the overall band structure is relatively simple near E_F , especially true for the $k_z \sim 0$ plane with the tunable Dirac gap. As a result, the DFT calculations provide a faithful description of the low-energy electron structure. The agreement between ARPES and DFT lays a foundation for future theoretical prediction and understanding of the underlying exotic quantum states in Fe₃Ge and its siblings. (ii) The Dirac fermion near E_F (DP1) can already be readily manipulated by the spontaneous spin reorientation of Fe₃Ge, thus, upon the application of an external magnetic field along the c axis, the DP1 gap size will be further enhanced and, accordingly, the upper branch of DP1 will be pushed closer to E_F . Such band modifications in Fe₃Ge could lead to a large Berry curvature near E_F , which has been suggested to be responsible for the novel quantum phenomena in kagome lattices, like the orbital magnetism and giant anomalous Hall effect.

Additionally, as stated in the last version, another main finding of our manuscript regarding the CDW order also offers the insightful information to understand the CDW instabilities in magnetic kagome systems. Thus, taken together, we believe that our work is commensurate with the high standards of Nature Communications. We have carefully revised the manuscript following the other suggestions and comments from Reviewer #2, please see our point-by-point response in the following.

Comment 1 (Previous comment 5): The revised manuscript still does not explain the significant difference in energy levels of DP2 observed in the calculated and experimental band structures. Band renormalization from electronic correlation may account for such a discrepancy. It may be necessary to employ many-body computations such as DMFT for a more comprehensive analysis of the band structure.

Reply: We appreciate Reviewer #2 for this comment. As discussed in the last round of review, the

concerned discrepancy is most likely caused by the moderate electron correlations in Fe₃Ge not being included in the DFT calculations. To further validate this point, we now directly compare the experimental data along the *H-L-H* direction with the corresponding DFT calculated band structure, which is renormalized by a factor of about 3 (Table S4). As shown in Fig. R3, the renormalized DFT calculations can provide a reasonable match with the low-energy ARPES spectra. In particular, one can identify a good correspondence between experiment and calculation in the energy positions of the DP2 and vHS2, as well as the dispersions connecting the DP2 and vHS2 (vHS3).

In our opinion, Reviewer #2's suggestion to perform the many-body DMFT calculations is a very good starting point for further investigating the intricate interplay of electron correlation effects and kagome-related physics in Fe₃Ge and its siblings. Such research will open pathways to explore more exotic topological phenomena and correlated quantum states. But this is beyond the scope of our current study.

In the revised manuscript, we have provided Fig. R3 as updated Fig. S14; we have also added the following sentence into page 16 of the main text: "The calculated band structure with renormalization can also clarify the aforementioned difference between experiment and calculation in the binding energy of DP2 (Supplementary Fig. S14)."

Figure R3 | ARPES intensity plot measured along the *H-L-H* direction with the photon energy of 116 eV (LH polarization). The figure is adopted from Fig. 2d. The red curves are DFT calculated band dispersions renormalized by a factor of about 3. The DP2, vHS2,3, and δ band are marked out.

Comment 2: The revised manuscript includes more discussions and Figure 5 regarding the CDW phase. In this section, by comparing between CsV₃Sb₅ and FeGe materials, the authors directly conclude on page 16 that "the vHS1 near EF arises mainly from the 3d_{xy}/3d_{x²-y²} and 3d_{z²} orbital characters (Fig. 4 and Supplementary Fig. S9), it thus would not induce the electronic instabilities in Fe₃Ge." It appears to imply that only vHS contributions from dxz/yz orbitals can lead to electronic instabilities in kagome materials. The authors should clarify this point further.

Reply: We thank Reviewer #2 for raising this concern. We find that our original phraseology used in the discussion of the vHS nesting (page 16 of the last version) can cause ambiguities. Following Reviewer #2's suggestion, here we state the main points of this part more clearly. Our results show that there exist two vHSs (vHS1 and vHS3) near E_F . The vHS1 in the $k_z \sim 0$ plane arises mainly from the 3d_{xy}/3d_{x²-y²} and 3d_{z²} orbital characters, while the vHS3 in the $k_z \sim \pi$ plane is primarily associated with the 3d_{xy}/3d_{x²-y²} and 3d_{xz}/3d_{yz} orbitals. According to the autocorrelation map in $k_z \sim 0$ (Fig. 5a), there is no nesting between the vHS1; in contrast, the vHS3 is nested by a wave vector of $(\pi, 0)$, as revealed by the autocorrelation map in $k_z \sim \pi$ (Fig. 5b). These observations suggest that the contribution of vHSs to the Fermi surface nesting

is most likely orbital dependent in Fe₃Ge, with those containing the 3d_{xz}/3d_{yz} orbitals being dominant. This is compatible with the recent findings in FeGe [*Nat. Phys.* **19**, 814-822 (2023)] and CsV₃Sb₅ [*Nat. Phys.* **18**, 301-308 (2022)] that the Fermi surface contours associated with the 3d_{xz}/3d_{yz} vHSs provide a better nesting condition.

To clarify this point, in the revised manuscript, we have rephrased the discussion of the vHS nesting.

Comment 3: With both vHS1 mainly from the 3d_{xy}/x²-y²/z² orbital characters, and vHS3 mainly from dxz/yz components near EF, Fe₃Ge does not exhibit CDW order. The main conclusion of this paper regarding the presence of orbital-selective vHSs near EF as the vital ingredient for triggering the CDW transition is quite confusing.

Reply: We appreciate Reviewer #2 for this comment. For clarity, here we explain in more detail how we reached this conclusion based on the current results. In the previous studies of FeGe, it has been proposed that both the orbital-selective vHSs near E_F (mainly 3d_{xz}/3d_{yz} orbitals) and the electron correlation effects should be taken into consideration to understand the origin of the CDW order. But so far, it is still quite unclear which one is the major underlying factor triggering the CDW transition in the magnetic kagome system. One of the goals of our work is to give crucial insights into this open question.

As discussed in the last version (main text, page 17, paragraph 1), although the nesting between the vHS3 is revealed around the $k_z \sim \pi$ plane, the 3D nature of Fe₃Ge could give rise to the vHS3 being near E_F only in a small range of k_z , rendering this in-plane nesting not sufficient to cause the charge fluctuations on the entire 3D Fermi surface. As a result, the nesting of vHSs in Fe₃Ge is not sufficiently strong to induce the electronic instabilities. This is in stark contrast to the FeGe case where there exists strong vHS nesting, as evidenced by the presence of both in-plane and out-of-plane vHS nesting and the opening of the CDW gap on vHS bands [*Nat. Phys.* **19**, 814-822 (2023); *Nature* **609**, 490-495 (2022)]. Given that the moderate electron correlations are present in both the non-charge-ordered Fe₃Ge and the charge-ordered FeGe while the nesting of vHSs is much stronger in the latter compound, it is thus most likely that the electronic instabilities induced by the strong vHS nesting is the dominant factor for the formation of CDW order in a magnetic kagome lattice.

To avoid the ambiguity, in the revised manuscript, we have rephrased the discussion of the CDW physics.

Reviewer #3 (Remarks to the Author):

I commend the Authors for their efforts in addressing all the reviewers' comments. All the concerns I expressed in my previous report have been satisfactorily addressed: a clear comparison between two- and one-Gaussian fit of the DP1 spectral feature is now presented and discussed in Fig.S8 and 'Supplementary Note 3', along with additional data acquired at higher temperature of 220K. I therefore recommend the manuscript for publication in Nature Communications.

Reply: We sincerely thank Reviewer #3 for carefully reviewing our revised manuscript and recommending its publication in Nature Communications.

List of changes to manuscript

- We added or revised the following figures:
 1. Fig. S6b-e to add the quantitative fitting of the EDC at K point and the EDCs near M point (K - M direction);
 2. Fig. S7a,b to add the evidence for the two band dispersions forming the DP1 along the Γ - K direction;
 3. Fig. S14 to add the direct comparison between ARPES spectra and renormalized DFT calculations (H - L - H direction).
- We added the following section into the Supplementary Information:

Note 3 - Observation of the two branches of DP1 along the Γ - K direction in the second Brillouin zone.
- Revisions to the text of the main manuscript (used a red font to mark out):
 1. Clarified our studies regarding the CDW order in the abstract, introduction, discussion, and conclusion;
 2. Clarified that the second branch of DP1 along the Γ - K direction is not visible in the first Brillouin zone but can be observed in the second Brillouin zone;
 3. Clarified the difference between experiment and calculation in the binding energy of DP2;
 4. Rephrased the discussion of the vHS nesting;
 5. Rephrased the discussion of the CDW physics.

Reviewer #1 (Remarks to the Author):

While resubmitted draft by R. Lou et al. of the "Orbital-selective effect of spin reorientation on the Dirac fermions in a non-charge-ordered kagome ferromagnet Fe₃Ge" answers majority of my questions, I would like a further clarification for my question 3 before I can recommend this article for publication in Nature Communications.

Reply: We sincerely thank Reviewer #1 for the careful review and the positive recommendation for the publication of our manuscript after revision. We have carefully revised the manuscript following their valuable comments.

While the authors explained and clarified well their assessment of an increased the k_z smearing at $k_z = \pi$ (116 eV, 146 eV) compared to $k_z = 0$ (125 eV), as well as reasons for (not) using $k_z = \pi/2$ data (135 eV), it is still not fully clear to me why the authors present in different figures measurements at different $k_z = \pi$ values. Namely, in figure 2b, we see data taken at 146 eV ($k_z = \pi$), while in the figure 2d we have data taken at $h\nu = 116$ eV, which is also $k_z = \pi$ based on the figure S4. However, 2D curvature and EDCs in figure 3 h,i, j and k are then taken at 146 eV. The authors state in their response (to my question 3) that the 116 eV data are in general more intense than 146 eV mappings, so I am wondering what is the reason that not all of the $k_z = \pi$ data is taken at this energy. In particular looking at the figure 2b and figure S4, 116 eV is indeed more intense and sharper, but this data is not used in the main text to show/indicate the Dirac pocket at H.

Reply: We appreciate Reviewer #1 for raising this concern. In the last round of review, our point is actually that, compared with the DFT calculated Fermi surfaces in $k_z = \pi$ (Fig. S5c,d), the additional spectral features in the 116 eV-mappings (Fig. S4b,c) are generally more intense than that in the 146-eV mappings (Fig. 2b) due to the k_z selection rules. As discussed before, these additional patterns are the projections from other k_z planes. Specifically, as seen in Fig. S4b, there is an additional Fermi pocket surrounded by the Dirac pocket at H point; and in Fig. 2d, additional Dirac-like bands close to the DP2 bands are observed around H point near E_F (see the positive momentum side). In contrast, these k_z -projected features close to the DP2 pocket in the 116-eV data are less visible in the 146-eV data, as seen from Figs. 2b and 3h. Therefore, to avoid the complex k_z -projected spectra and unambiguously investigate the DP2-related spectral features, in the main text, instead of 116 eV, we used 146-eV photons for most of the measurements in the $k_z \sim \pi$ plane (Figs. 2b, 3, and 5).

Figures adopted from Figs. S4b and 2d, respectively. The blue arrows indicate the k_z -projected features close to the DP2 pocket.

On the other hand, in Fig. 2d, we presented the 116-eV data for two reasons: one is to show the dispersion connecting the DP2 and vHS2, which is much clearer than that in the 146-eV data; and the other is, more importantly, to compare with the 135-eV data in Fig. 2e, their resemblance is a strong indication that the ARPES intensity in the $k_z \sim \pi$ plane suffers from a stronger k_z broadening effect than the $k_z \sim 0$ plane, as discussed before.

To clarify this point, we have added the following sentences into the captions of Fig. 2b,d.

For Fig. 2b: “To avoid the complex k_z -projected spectra (see Supplementary Fig. S4b,c), the 116-eV photons were not utilized to study the DP2-related spectral features here and hereafter, except for the measurements in **d**, which will be explained later.”

For Fig. 2d: “We presented the 116-eV data in **d** for two reasons: one is to show the dispersion connecting the DP2 and vHS2, which is much clearer than that in the 146-eV data; and the other is, more importantly, to compare with the 135-eV data in **e**.”

Reviewer #2 (Remarks to the Author):

I read carefully the revised manuscript as well as authors' replies, and found that most of my questions and concerns are answered. Considering the great efforts made by authors in two-round reviewing, I would give my support for publication of the revised form in NC.

Reply: We sincerely thank Reviewer #2 for carefully reviewing our revised manuscript and recommending its publication in Nature Communications.

List of changes to manuscript

- Revisions to the text of the main manuscript (used a red font to mark out):
 1. Caption of Fig. 2b, added “To avoid the complex k_z -projected spectra (see Supplementary Fig. S4b,c), the 116-eV photons were not utilized to study the DP2-related spectral features here and hereafter, except for the measurements in **d**, which will be explained later.”
 2. Caption of Fig. 2d, added “We presented the 116-eV data in **d** for two reasons: one is to show the dispersion connecting the DP2 and vHS2, which is much clearer than that in the 146-eV data; and the other is, more importantly, to compare with the 135-eV data in **e**.”